# Latent Exploration for Reinforcement Learning

**Alberto Silvio Chiappa**
École Polytechnique Fédérale de Lausanne (EPFL)
`alberto.chiappa@epfl.ch`

**Alessandro Marin Vargas**
EPFL
`alessandro.marinvargas@epfl.ch`

**Ann Zixiang Huang**
Mila, EPFL
`zixiang.huang@mail.mcgill.ca`

**Alexander Mathis**
EPFL
`alexander.mathis@epfl.ch`

## Abstract

In Reinforcement Learning, agents learn policies by exploring and interacting with the environment. Due to the curse of dimensionality, learning policies that map high-dimensional sensory input to motor output is particularly challenging. During training, state of the art methods (SAC, PPO, etc.) explore the environment by perturbing the actuation with independent Gaussian noise. While this unstructured exploration has proven successful in numerous tasks, it can be suboptimal for overactuated systems. When multiple actuators, such as motors or muscles, drive behavior, uncorrelated perturbations risk diminishing each other's effect, or modifying the behavior in a task-irrelevant way. While solutions to introduce time correlation across action perturbations exist, introducing correlation across actuators has been largely ignored. Here, we propose LATent TIme-Correlated Exploration (Lattice), a method to inject temporally-correlated noise into the latent state of the policy network, which can be seamlessly integrated with on- and off-policy algorithms. We demonstrate that the noisy actions generated by perturbing the network's activations can be modeled as a multivariate Gaussian distribution with a full covariance matrix. In the PyBullet locomotion tasks, Lattice-SAC achieves state of the art results, and reaches 18% higher reward than unstructured exploration in the Humanoid environment. In the musculoskeletal control environments of MyoSuite, Lattice-PPO achieves higher reward in most reaching and object manipulation tasks, while also finding more energy-efficient policies with reductions of 20-60%. Overall, we demonstrate the effectiveness of structured action noise in time and actuator space for complex motor control tasks. The code is available at: https://github.com/amathislab/lattice.

## 1   Introduction

Effectively exploring the environment while learning the policy is key to the success of Reinforcement Learning (RL) algorithms. Typically, exploration is attained by using a non-deterministic policy to collect experience, so that the random component of the action selection process allows the agent to visit states that would have not been reached with a deterministic policy. In on-policy algorithms, such as A3C [1] and PPO [2], the policy network parametrizes a probability distribution, usually an independent multivariate Gaussian, from which actions are sampled. In off-policy algorithms, such as DQN [3], DDPG [4] and SAC [5], the policy used to collect experience can be different from the one learned to maximize the cumulative reward, which leaves more freedom in the selection of an exploration strategy. The standard approach in off-policy RL consists in perturbing the policy by introducing randomness (e.g., $\epsilon$-greedy exploration in environments with a discrete action space and Gaussian noise in environments with a continuous action space). In the past years, many notable

37th Conference on Neural Information Processing Systems (NeurIPS 2023).

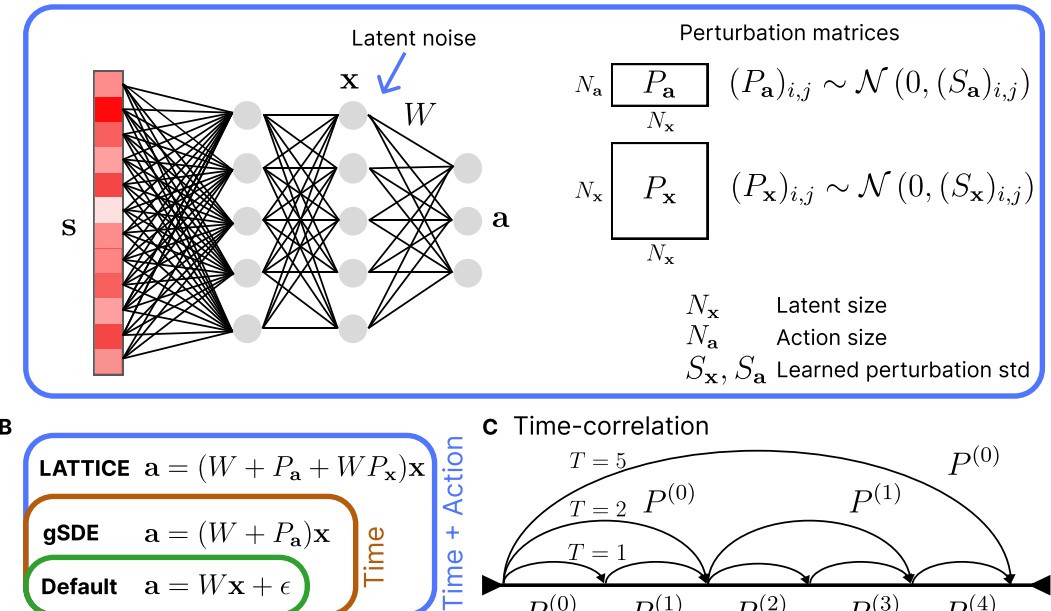

Figure 1: **A** Lattice introduces state-dependent perturbations both to the action space and to the latent space. **B** Compared to the default action noise, Generalized State-Dependent Exploration (gSDE) [8] introduces a state-dependent perturbation of the action. Lattice generalizes gSDE, including a second, policy-dependent perturbation, which induces correlation between action components. **C** Similarly to gSDE, the perturbation matrices $P_{\mathbf{x}}$ and $P_{\mathbf{a}}$ are sampled periodically, providing noise with a temporal structure.

successes of RL are based on independent noise [1, 2, 4, 6, 7]. However, previous work has challenged the inefficiency of unstructured exploration by focusing on introducing temporal correlations [8, 4, 9–11]. Indeed, two opposite perturbations of subsequent actions might cancel out the deviation from the maximum probability trajectory defined by the policy, effectively hiding potentially better actions if followed by a more coherent policy. Here, we argue that not only correlation in time [4, 9–11, 8], but also correlation across actuators, can improve noise-driven exploration.

In motor control, precise coordination between actuators is crucial for the execution of complex behaviors [12–15]. Perturbing each actuator independently can disrupt such coordination, limiting the probability of discovering improvements to the current policy. In particular, musculoskeletal systems [16–19] typically feature a larger number of actuators (muscles) than degrees of freedom (joints). Here, we demonstrate that in such systems the exploration achieved by uncorrelated actuator noise is suboptimal. To tackle this problem, we introduce an exploration strategy, named LATent TIme-Correlated Exploration (Lattice). Lattice takes advantage of the synergies between actuators learned by the policy network to perturb the different action components in a structured way. This is achieved by applying independent noise to the latent state of the policy network (Fig. 1 A). With extensive experiments, we show that Lattice can replace standard unstructured exploration [2, 5] and time-only-correlated exploration (gSDE) [8] in off-policy (SAC) and on-policy (PPO) RL algorithms, and improve performance in complex motor control tasks. Importantly, we demonstrate that Lattice-SAC is competitive in standard benchmarks for continuous control, such as the locomotion environments of PyBullet [20]. In the Humanoid locomotion task, Lattice improves training efficiency, final performance and energy consumption. In the complex object manipulation tasks of the MyoSuite library [18], involving the control of a realistic musculoskeletal model of the human hand, Lattice-PPO achieves higher reward, while also finding more energy-efficient policies than standard PPO. To the best of our knowledge, our work is the first to showcase the potential of modeling the action distribution of a policy network with a multivariate Gaussian distribution with full covariance matrix induced by the policy network weights, rather than a diagonal covariance matrix.

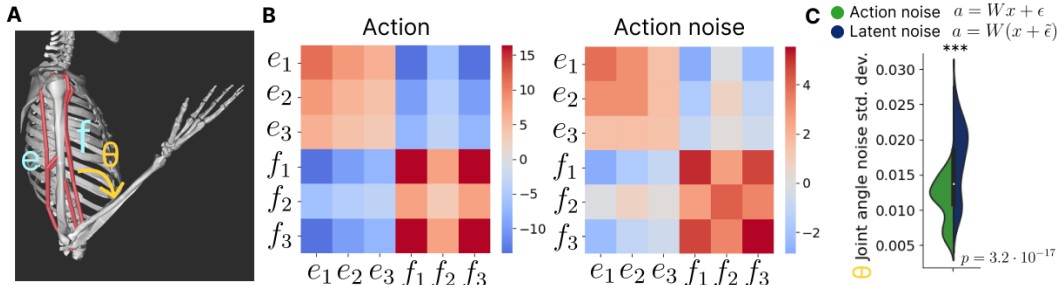

Figure 2: **A** Representation of a musculoskeletal model of a human arm. The elbow joint is actuated by flexor-extensor muscle groups. Figure adapted from [18, 31]. **B** Covariance matrix of the actions (left) and noise of the actions (right) when latent state of a learned policy is perturbed for flexors $f_1$-$f_3$ and extensors $e_1$-$e_3$. **C** Distribution of the joint angle perturbation across episodes when noise is applied to the action or to the latent space. The action noise was tuned so that each action component has the same variance as with latent noise. The difference in distribution, due to the extra-diagonal terms of the action covariance, is statistically significant ($p = 3.2 \times 10^{-17}$, Wilcoxon signed-rank test).

## 2 Related work

**Exploration based on the learned policy.** In on-policy RL, the target policy must be stochastic, so that the experience collected includes enough variability for the agent to discover policy improvements. Algorithms such as A3C [1], PPO [2], and TRPO [21] parametrize each action component with an independent Gaussian distribution (continuous actions) or a categorical distribution (discrete actions). In off-policy RL, most exploration strategies are based on perturbations of the policy. For example, in discrete action spaces an exploratory policy can be derived from the target policy by adding $\epsilon$-greedy exploration, or more sophisticated perturbations such as the upper confidence bound [22, 23], Thompson sampling [24], Boltzmann exploration [25] and Information-Directed Sampling (IDS) [26]. In continuous action space, the perturbation of the actions is provided by independent random noise, with (e.g., SAC [5]) or without (e.g., DDPG [4]) learned noise parameters. Unstructured exploration has several drawbacks [9], which have been tackled by ensuring a temporally coherent noise through colored noise [11], an auto-regressive process [27] or a Markov chain [28]. Another strategy to induce time correlation consists in keeping the perturbation parameters constant for multiple steps. This concept has been applied to perturbations of the network parameters [29] and to state-dependent action perturbations [9, 10, 8, 30]. While these exploration strategies provide smoother noise in time, Lattice aims to pair time-correlated state-dependent exploration with correlation in the action space.

**Other exploration strategies.** In on-policy RL, count-based exploration [32], curiosity-driven exploration [33] and random network distillation [34] are techniques to encourage visiting rare states. In off-policy RL, bootstrapped DQN [35] employs multiple Q-networks to guide exploration and improve performance. Additionally, unsupervised methods like Diversity Is All You Need [36] can achieve more diverse exploration by maximizing mutual information between a goal-conditioned policy and a set of skill labels. Starting to explore only after returning to a recently visited state allows to solve complex sequential-decision problem [37]. Lattice does not aim to replace curriculum learning [38–40, 15] or exploratory policies, but proposes exploring implicitly via the policy network in a monolithic way and can be combined with other approaches.

**Exploration in over-actuated system.** Overactuated systems, such as musculoskeletal systems, pose a challenge for exploration. Recent NeurIPS challenges have addressed the problem of learning control policies for locomotion and dexterous hand movements with musculoskeletal models [41, 14, 15]. The winners in these challenges have typically utilized complex training curricula, such as static-to-dynamic stabilization in the Baoding task, where a hand needs to accurately rotate two balls [15], or feature engineering and expert demonstration for locomotion [41]. Exploration can also be encouraged by learning a muscle-coordination network [42] or the state-dependent joint torque limits and energy function [43]. Similarly, Schumacher et al. proposed a self-organizing controller that outperforms previous approaches by inducing state-space covering exploration [19]. Integrating exploratory policies into a RL framework requires ad-hoc solutions, whereas Lattice can be trained end-to-end with both on-policy and off-policy algorithms.

# 3 Motivating example: The case of a flexor-extensor, single joint arm

To motivate Lattice, consider a simple flexor-extensor system, such as the elbow joint in the human upper arm (Fig. 2A). The activation $a_f \in [0, 1]$ of the flexor muscle (biceps) produces a negative angular acceleration at the elbow, while the activation $a_e \in [0, 1]$ of the extensor muscle (triceps) produces a positive angular acceleration. Using a first order approximation of the system dynamics, we can write the equation of the angular position of the elbow as $\ddot{\theta} = \alpha(a_e - a_r)$, where $\theta$ is the angle between the upper arm and the lower arm and $\alpha \, [\mathrm{rad\,s^{-2}}]$ is a parameter of the system which converts the muscle activations into angular acceleration. Given this system, we consider a simple control problem, where an agent needs to reach a target angular position of the elbow by activating the flexor and/or the extensor muscle. At every step, the agent observes the difference $\Delta\theta = \theta_0 - \theta_t$ between the target angle $\theta_0$ and the angle $\theta_t$ at time $t$. For small values of $\Delta\theta$, we assume that the control policy learned by the agent can be approximated with the linear functions $a_e = 0.5 + \Delta\theta$ and $a_f = 0.5 - \Delta\theta$. We now want to compare the effects on the angular acceleration produced by noise applied to the actions ($a_e$ and $a_f$) or to the latent state ($\Delta\theta$) for various cases:

**Case 1: action space noise.** If we inject independent noise $\epsilon_f$ and $\epsilon_e$ following a normal distribution $\mathcal{N}(0, \sigma^2)$ to the muscle activations, we have $a_f \sim \mathcal{N}(\langle a_f \rangle, \sigma^2)$ and $a_e \sim \mathcal{N}(\langle a_e \rangle, \sigma^2)$. It can be shown that $\ddot{\theta} \sim \mathcal{N}(\alpha(\langle a_e \rangle - \langle a_f \rangle), 2\alpha^2\sigma^2)$ (Appendix A.1).

**Case 2: latent space noise.** If we instead inject Gaussian noise to the latent state $\Delta\theta$, while the marginal distribution of $a_f$ and $a_e$ is the same as with action space noise, the fact that the two distributions are correlated has an effect on the distribution of $\ddot{\theta}$. It can be shown that $\ddot{\theta} \sim \mathcal{N}(\alpha(\langle a_e \rangle - \langle a_r \rangle), 4\alpha^2\sigma^2)$ (Appendix A.1).

This toy example illustrates how perturbing the latent state, instead of the actions, leads to higher variance in the *behavior* space. In fact, when randomly perturbing two opposing muscles, half of the time the perturbations will be in opposite directions, with a reduced impact on the observed kinematics.

To corroborate this finding about the advantage of latent versus action perturbations, we considered a policy network trained to control a realistic musculoskeletal arm model, implemented in MyoSuite [18], featuring three flexor muscles and three extensor muscles. A neural network trained with PPO and Lattice exploration can learn to accurately reach a target angle in every episode, and while doing so learns to alternately activate the extensor or the flexor muscle group (Fig. 2 B, left). The training procedure is detailed in Appendix A.1. We collected a dataset of 100 episodes where the latent state of the policy network was perturbed with Gaussian noise. While applying independent random noise to the muscle activations produces a diagonal covariance matrix, perturbing the latent state of the network leads to a full covariance matrix (Fig. 2 B, right). Thus, as hypothesized, the last layer of the policy network learned to distinguish between agonist and antagonist muscles, namely, the covariance shows positive correlation among muscles of the same group. We then tested whether sampling action noise with this covariance matrix leads to higher variance in the task space (joint angles) than noise sampled independently for each muscle. We collected 100 episodes where, at each step, we computed an action with action space perturbation and one action with latent state perturbation. For consistency of evaluation, the perturbation in the action space was sampled with the same variance magnitude induced by the latent perturbation. We executed each pair of actions in two cloned environments and compared the variance of the joint angles. The latent state perturbation introduces higher variance in the joint angles (Fig. 2 C), thus driving more diverse kinematics and wider exploration. Overall, we conclude that latent exploration might be advantageous when learning to control over-actuated systems, and thus developed and empirically tested Lattice on different benchmarks.

# 4 Methods

## 4.1 LATent TIme-Correlated Exploration (Lattice)

In continuous control, the policy update of on-policy algorithms (e.g., REINFORCE [44], A3C [1], TRPO [21], PPO [2]) and of some off-policy algorithms (e.g., SAC [5]) requires the computation of $\log \pi(a_t|s_t)$, which is the logarithm of the probability of choosing a certain action $a_t$ given the current state $s_t$, according to the current policy $\pi$. In such cases, the policy gradient needs to be

| **Algorithm 1** Standard (e.g., PPO, SAC) | **Algorithm 2** Lattice |
|---|---|

**Require:**

Policy $\pi$, env. dynamics $p$,

std array $\boldsymbol{\sigma}^{(\mathbf{a})}$, $(\boldsymbol{\epsilon_a})_i \sim \mathcal{N}(0, \sigma_i^{(\mathbf{a})})$

1: Initialize state $\mathbf{s}_t$;
2: step_count $\leftarrow 0$;
3: **while** not done **do**
4:     $\mathbf{x}_t \leftarrow \pi(\mathbf{s}_t)$;
5:     $\boldsymbol{\epsilon_a} \leftarrow$ sample $(\boldsymbol{\sigma}^{(\mathbf{a})})$;
6:     $\mathbf{a}_t \leftarrow W\mathbf{x}_t + \boldsymbol{\epsilon_a}$;
7:     $\mathbf{s}_{t+1}, r_t,$ done $\leftarrow p(\mathbf{s}_t, \mathbf{a}_t)$;
8:     step_count $\leftarrow$ step_count $+1$;
9: **end while**

**Require:**

Policy $\pi$, env. dynamics $p$,

period $T$, matrix $S^{(\mathbf{a})}$, matrix $S^{(\mathbf{x})}$, $\alpha \in \{0, 1\}$

$(P_{\mathbf{a}})_{i,j} \sim \mathcal{N}(0, S_{i,j}^{(\mathbf{a})})$, $(P_{\mathbf{x}})_{i,j} \sim \mathcal{N}(0, S_{i,j}^{(\mathbf{x})})$

1: Initialize state $\mathbf{s}_t$;
2: step_count $\leftarrow 0$;
3: **while** not done **do**
4:     **if** step_count mod $T = 0$ **then**
5:         $P_{\mathbf{a}} \leftarrow$ sample $(S^{(\mathbf{a})})$;
6:         $P_{\mathbf{x}} \leftarrow$ sample $(S^{(\mathbf{x})})$;
7:     **end if**
8:     $\mathbf{x}_t \leftarrow \pi(\mathbf{s}_t)$;
9:     $\mathbf{a}_t \leftarrow (W + P_{\mathbf{a}} + \alpha W P_{\mathbf{x}})\, \mathbf{x}_t$;
10:     $\mathbf{s}_{t+1}, r_t,$ done $\leftarrow p(\mathbf{s}_t, \mathbf{a}_t)$;
11:     step_count $\leftarrow$ step_count $+1$;
12: **end while**

Algorithms 1-2: Experience collection algorithm with independent Gaussian noise (left) and with time- and actuator-correlated noise (right). In green, we highlight the parameters defining the independent noise. In brown, the elements inducing time-correlation (gSDE). In blue, those implementing actuator correlation through the perturbation of the latent state (Lattice). Note that for $\alpha = 0$, Lattice is equivalent to gSDE.

backpropagated through the density function of the action distribution. This is typically accomplished via the *reparametrization trick*, in which the policy network outputs the parameters of a differentiable probability density function, whose analytical expression is known, so that gradients can flow through the probability estimation. In the standard case, when we apply independent Gaussian noise to the action components, the action probability distribution can be parametrized as a multivariate Gaussian with diagonal covariance matrix (Algorithm 1). We propose that the weights of the policy network can be naturally used to parameterize the amount of exploration noise (Algorithm 2).

While a generic perturbation of the policy network's activations would lead to an action distribution with an unknown probability density function, in Lattice we limit the latent perturbation to the last layer's latent state, which is linearly transformed into the action (Fig. 1 A). Consider the output of the last layer of the policy network $\mathbf{x} \in \mathbb{R}^{N_x}$ and the matrix $W \in \mathbb{R}^{N_a \times N_x}$, mapping the embedding state to an action according to $\mathbf{a} = W\mathbf{x}$. We have indicated the size of the latent space with $N_x$ and the size of the action space with $N_a$. If we assume a perturbation of the latent state $\mathbf{x}$ with independent Gaussian noise $\boldsymbol{\epsilon} = \mathcal{N}(\mathbf{0}, \Sigma_{\mathbf{x}})$, where $\Sigma_{\mathbf{x}}$ is a positive-definite diagonal matrix, then the perturbed latent state $\tilde{\mathbf{x}}$ is distributed as $\tilde{\mathbf{x}} \sim \mathcal{N}(\mathbf{x}, \Sigma_{\mathbf{x}})$. Thus, the action distribution $\mathbf{a} \sim \mathcal{N}(W\mathbf{x}, W\Sigma_{\mathbf{x}}W^\top + \Sigma_{\mathbf{a}})$, where with $\Sigma_{\mathbf{a}}$ we have indicated the independent component of the action noise, can be derived as a linear transformation of a multivariate Gaussian distribution (details in Appendix A.2). This formula provides an analytical expression for $\log \pi(\mathbf{a}_t|\mathbf{s}_t)$, as a function of the policy network weights and the covariance matrix of the latent state:

$$\log \pi(\mathbf{a}|\mathbf{s}) = -\frac{N_a}{2}\log(2\pi) - \frac{1}{2}\log|W\Sigma_{\mathbf{x}}W^\top + \Sigma_{\mathbf{a}}| - \frac{1}{2}(W\mathbf{x} - \mathbf{a})^\top (W\Sigma_{\mathbf{x}}W^\top + \Sigma_{\mathbf{a}})(W\mathbf{x} - \mathbf{a}).$$
(1)

The probability of an action depends on the network weights both through its mean value $W\mathbf{x}$ and through the perturbation variance matrices $\Sigma_{\mathbf{x}}$ and $\Sigma_{\mathbf{a}}$. Depending on the RL algorithm, it might be convenient not to propagate the gradients of the policy network through the variance component of the loss. In this way, the expected action trains the policy network, while the loss due to the standard deviation can regulate its magnitude by updating the parameters $\Sigma_{\mathbf{x}}$ and $\Sigma_{\mathbf{a}}$.

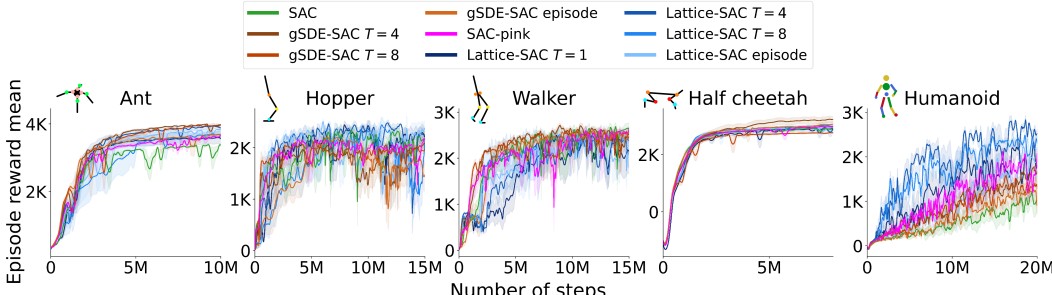

Figure 3: Learning curves in five locomotion tasks of PyBullet, representing the cumulative reward of each training episode (mean ± s.e.m. across random seeds). Our baseline results improve over the benchmark rewards of the pre-trained networks of RL Zoo [46], possibly because we used 16 vectorized environments to collect transitions and therefore a higher number of steps. The period $T$ refers to the time correlation of the exploration noise. Lattice, gSDE and Pink Noise perform comparably in all environments apart from the Humanoid [16], where Lattice is more efficient and achieves higher average reward.

## 4.2 Lattice generalizes time-correlated noise

Lattice can be thought of as an extension of Generalized State-Dependent Exploration (gSDE) [8], as it inherits all its properties, while extending it with the possibility of perturbing the latent state of the policy, besides the actions (Fig. 1 B). In Lattice, the agent learns two sets of parameters, $S_\mathbf{x}$ and $S_\mathbf{a}$, representing the standard deviation of $N_\mathbf{x} \times (N_\mathbf{x} + N_\mathbf{a})$ Gaussian distributions ($N_\mathbf{x}$ and $N_\mathbf{a}$ being the size of the latent space and of the action space, respectively). These parameters are used to sample, every $T$ steps, two noise matrices $P_\mathbf{x}$ and $P_\mathbf{a}$ (Fig. 1 C). These matrices are used to determine the latent noise $\boldsymbol{\epsilon}_\mathbf{x} = \alpha P_\mathbf{x}\mathbf{x}$ and the action noise $\boldsymbol{\epsilon}_\mathbf{a} = P_\mathbf{a}\mathbf{x}$. The parameter $\alpha \in \{0, 1\}$ can be used to turn on or off the correlated action noise. For $\alpha = 0$, Lattice is equivalent to gSDE. The value of $T$ regulates the amount of time-correlation of the noise. If $T$ is large, $P_\mathbf{x}$ and $P_\mathbf{a}$ are sampled infrequently, causing the noise applied to the latent state and to the action to be strongly time-correlated (Algorithm 2).

In short, the action distribution depends on the latent state $\mathbf{x}$, the last linear layer parameters $W$, the Lattice noise parameters $S_\mathbf{x}$ and the gSDE noise parameters $S_\mathbf{a}$ in the following way:

$$\pi(\mathbf{a}|\mathbf{s}) = \mathcal{N}\left(W\mathbf{x}, \text{Diag}(S_\mathbf{a}^2\mathbf{x}^2) + \alpha^2 W\text{Diag}(S_\mathbf{x}^2\mathbf{x}^2)W^\top\right) \tag{2}$$

## 4.3 Implementation details of Lattice

Code to study the implementation details is available at https://github.com/amathislab/lattice.

**Noise magnitude.** The magnitude of the variance of individual noise components can either be learned or kept fixed, and can or can not be made dependent on the current state. We extend the implementation of gSDE [8, 45], which makes the noise magnitude a learnable parameter independent of the policy network.

**Controlling the variance.** We introduce two clipping parameters to limit the minimum and the maximum value of the variance of each latent noise component, to avoid excessive perturbations and convergence to a deterministic policy. This clipping does not modify the analytical expression of the action distribution, because it does not affect the sampled values, but rather the variance of the distribution.

**Layer-dependent noise rescaling.** As the noise vectors are given by $\boldsymbol{\epsilon}_\mathbf{x} = \alpha P_\mathbf{x}\mathbf{x}$ and $\boldsymbol{\epsilon}_\mathbf{a} = P_\mathbf{a}\mathbf{x}$, assuming the components of $\mathbf{x}$ have similar magnitude, the noise scales with the size of the latent state. To remove this effect, we apply a correction to the learned matrices $\log \tilde{S}_\mathbf{x}$ and $\log \tilde{S}_\mathbf{a}$ before sampling $P_\mathbf{x}$ and $P_\mathbf{a}$: $\log S_\mathbf{x} = \log \tilde{S}_\mathbf{x} - 0.5 \log(N_\mathbf{x})$ and $\log S_\mathbf{a} = \log \tilde{S}_\mathbf{a} - 0.5 \log(N_\mathbf{x})$. It can be proven that this correction removes the dependence of the noise from the size of the latent state (Appendix A.3), and thus from the network architecture.

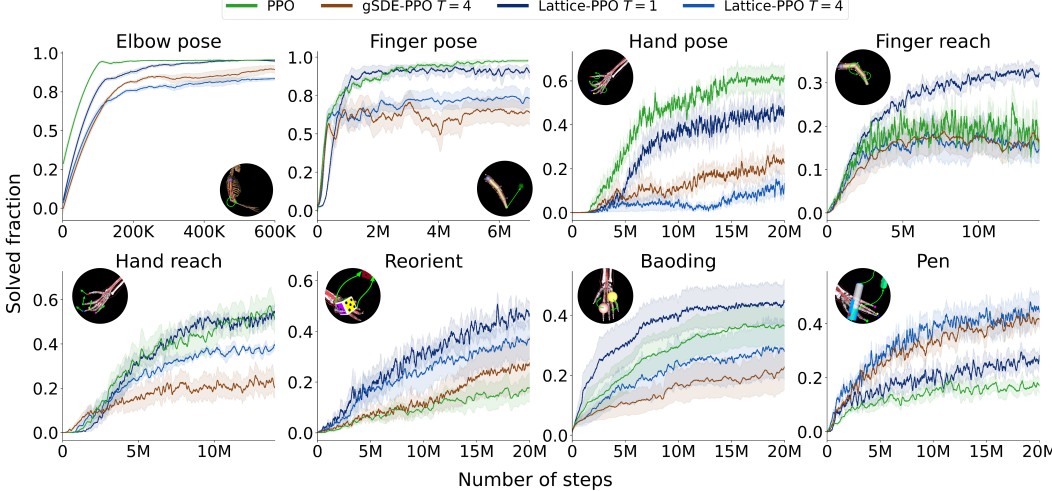

Figure 4: Learning curves in the MyoSuite environments, showing the adjusted solved fraction throughout the training (ratio between the number of steps in which the target is reached and the maximum length of the episode, mean ± s.e.m. across random seeds). In the pose environments, where the target prescribes the angle of all joints, standard exploration can be better than Lattice. In the other environments, where the exploration can leverage the learned muscle synergies, Lattice is on par with or better than independent noise. We omit the learning curves with time correlation above 4, as they were always worse or much worse than 1 and 4 in PPO, in accordance with previous findings for locomotion [8].

**Preventing singularity of the covariance matrix.** The covariance matrix of the action distribution must be positive definite. It has the expression $W\Sigma_{\mathbf{x}}W^\top + \Sigma_{\mathbf{a}}$ (Eq. 2), with $\Sigma_{\mathbf{x}} = \alpha^2\text{Diag}(S_{\mathbf{x}}^2\mathbf{x}^2)$ and $\Sigma_{\mathbf{a}} = \text{Diag}(S_{\mathbf{a}}^2\mathbf{x}^2)$, where the square operations are to be intended element-wise. By construction, the covariance matrix is positive-semidefinite. However, it could be singular, e.g., when $\mathbf{x}$ is the null vector (Appendix A.4 for further details). To prevent the singularity of the covariance matrix, we add a small positive regularization value to its diagonal terms.

## 5 Experiments

We benchmarked Lattice on standard locomotion tasks [47, 6, 16, 48–50] in PyBullet [20], as well as musculoskeletal control tasks of MyoSuite [18] built in MuJoCo [31]. Both libraries include continuous control tasks of varying complexity. While in PyBullet the actuators apply a torque to each individual joint, in MyoSuite the agent controls its body through muscle activations. We give a complete description of each task in Appendix A.6. We implemented Lattice as an extension of gSDE in the RL library Stable Baselines 3 [45], which we used for all our experiments; the hyperparameters of the algorithms are detailed in Appendix A.7. All the results are averaged across 5 random seeds. The training was run on a GPU cluster, for a total of approximately 10,000 GPU-hours.

### 5.1 Pybullet locomotion environments

Lattice can be paired with off-policy RL algorithms, such as SAC [5]. We tested this combination in the locomotion environments of PyBullet, where gSDE-SAC achieves state-of-the-art performance [8, 46]. We also included Pink Noise exploration [11] as an additional baseline, which is the state-of-the-art colored noise process for exploration with off-policy RL algorithms. We used the same network architecture and hyperparameters for SAC specified in [46] for all the environments (see Appendix A.7). Preliminary experiments on the parameters of Lattice showed that environments with a smaller action space benefit from a higher initial standard deviation of the exploration matrix, so we set all the elements of $\log S_{\mathbf{x}}$ and $\log S_{\mathbf{a}}$ to 0 for Ant and Humanoid and to 1 for the other environments. We found that Lattice achieves similar performance to gSDE-SAC and SAC-pink for lower-dimensional morphologies (Ant, Hopper, Walker and Half Cheetah), while outperforming them substantially for the Humanoid (Fig. 3 and Appendix A.8). This suggests that Lattice can achieve

higher performance when controlling larger-dimensional morphologies. Furthermore, we later show that the policy learnt with Lattice-SAC for the Humanoid is more energy efficient than that learnt with SAC (Section 6).

## 5.2 Musculoskeletal control: MyoSuite environments

For musculoskeletal control, we tested several tasks from the MyoSuite [18]:

- Three *pose* tasks (Elbow Pose, Finger Pose and Hand Pose). In each session, a target angular position is sampled independently for each joint. This means that the policy has to learn how to control each degree of freedom independently.
- Two *reach* tasks (Finger Reach and Hand Reach). In each session, a target position for each finger tip (one for Finger Reach and five for Hand Reach) is sampled independently.
- Three *object manipulation* tasks (Reorient, Pend and Baoding). In each session, the target is a fixed or moving target for an object (a die in Reorient, a pen in Pen and two Baoding balls in Baoding).

We tested Lattice with a LSTM network and PPO. This choice is motivated by (our) winning solution of the Boading ball task in the 2022 NeurIPS MyoChallenge [15], which used PPO together with a LSTM network [51]. For our experiments we use the same PPO hyperparameters and network architecture, which we keep identical across methods (see Appendix A.7).

We set the initial value of all the elements of $\log S_{\mathbf{x}}$ and $\log S_{\mathbf{a}}$ to 0 in every task, except in Elbow pose and Hand pose, where we set it to 1.

In the *reach* and challenging *object manipulation* tasks (Reorient, Baoding and Pen), where the policy network has to control a complex hand model with 39 muscles and receives an observation with more than 100 dimensions, Lattice consistently outperforms the baselines (Fig. 4 and Appendix A.8). In contrast, in the *pose* tasks PPO performs comparably or better than Lattice-PPO (Fig. 4 and Appendix A.8). Those tasks define dense target states and, perhaps, exploration and coordination is less important (see below).

Interestingly, focusing action noise in the task-relevant space allows Lattice to avoid activating muscles unnecessarily, leading to conspicuous energy saving in the manipulation and reaching tasks (Fig. 5). In *reach* and *object manipulation*, Lattice-PPO achieves better reward at lower energy cost. We speculate that injecting correlated noise across muscle activations improves exploration in the space of task-relevant body poses and facilitates the discovery of efficient, coordinated movements. We tested this hypothesis next.

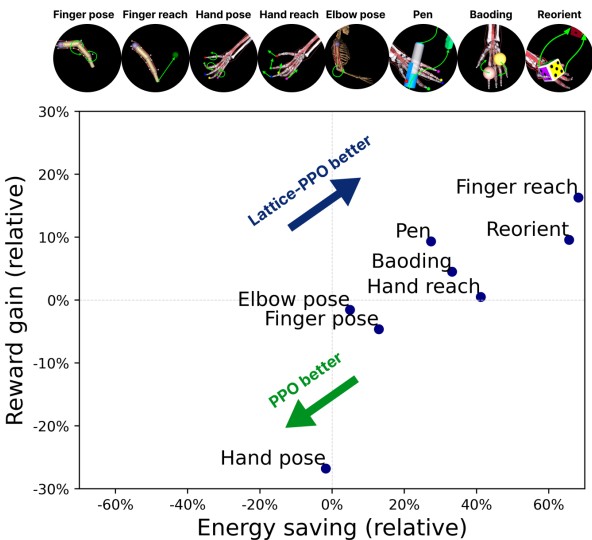

Figure 5: Energy saving versus reward gain for the muscle control tasks. Lattice learns policies 20% to 60% more energy efficient than independent exploration in the *reaching* and *object manipulation* tasks, while also achieving higher reward. In the *pose* tasks, independent exploration performs better, with similar energy consumption.

## 6 How does Lattice explore?

In most of the considered mukuloskeletal control environments, Lattice finds a more energy efficient control strategy, without compromising the performance (Fig. 5 and Appendix A.8). Especially in Reorient, where Lattice outperforms standard exploration in cumulative reward, it does so at a

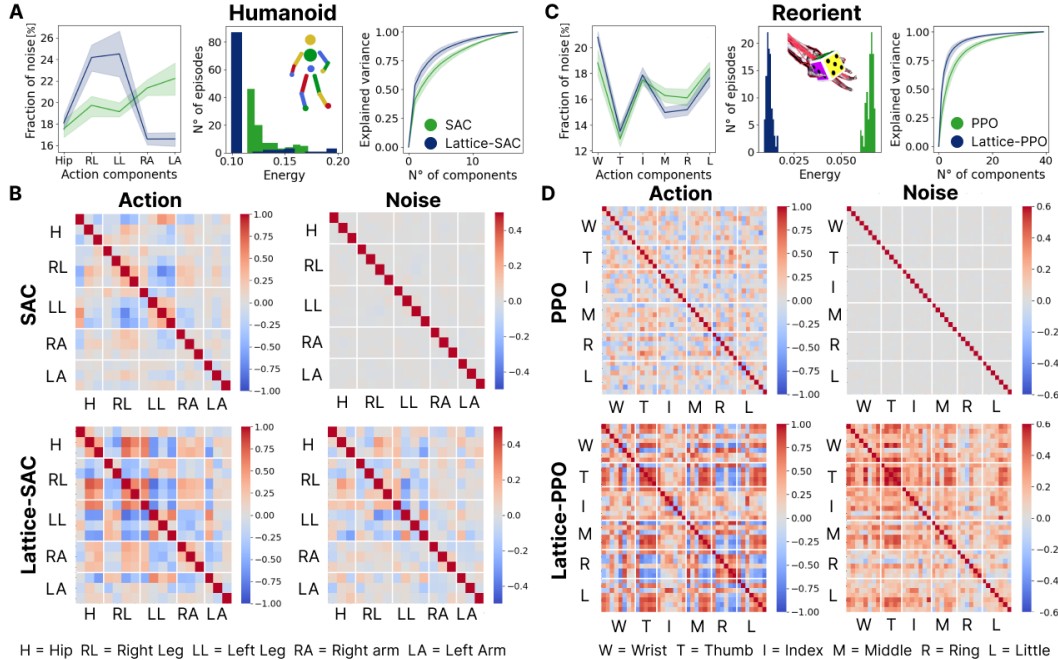

Figure 6: **A** Left: Graph of the fraction of noise allocated to each group of action components by a stochastic policy trained in the Humanoid environment with SAC and Lattice-SAC. In the Humanoid, Lattice tends to focus most of the variance on the task-relevant actuators (legs). Middle: Distribution of the energy consumption. Each bar represents the number of test episodes falling in the corresponding energy consumption interval. Right: Cumulative explained variance of the actions' principal components, computed from a dataset of 100 test episodes per seed. Shaded area represents 95% confidence interval across training seeds. More principal components are required to explain the same fraction of variance in SAC and PPO versus Lattice-SAC and Lattice-PPO. **B** Heatmap of the correlation matrix for the action space (left column) and noise of the action (right column) for SAC (top row) and Lattice-SAC (bottom row). **C-D** Same as A-B but analyzing a policy trained on the MyoSuite Reorient task with PPO and Lattice-PPO. Lattice tends to induce sharper correlation between action components. Local patterns of correlation between action components can be recognized in the noise covariance matrix.

fraction of the energy cost. We hypothesize that this is a direct consequence of biasing the noise with the same correlation as the actuators, preventing energy-consuming co-activations, which the agent needs to compensate for, despite the reward not being affected by them.

We assessed whether coordination emerged by analyzing policies for the Humanoid locomotion task and the Reorient task. By looking at the relative allocation of action noise across actuators (Fig. 6 A, C), we can see that in the Humanoid, compared to SAC, Lattice-SAC re-directs more exploration noise towards the task-relevant leg motors (SAC: 40% Legs , 45% Arms and 15% Body; Lattice-SAC: 50% Legs , 32% Arms, 18% Body). In Reorient, where all the muscles contribute to controlling the hand, the difference is less evident. Furthermore, both in the Humanoid and Reorient task, the intrinsic dimensionality of the actions output by a policy trained with Lattice is lower. Indeed, consistently across random seeds, fewer principal components explain a higher fraction of the action variance in Lattice than in standard exploration (Fig. 6 A, C). Lattice consistently promotes policies with actions lying on a lower-dimensional manifold, and this effect extends beyond Humanoid and Reorient, to almost all the considered tasks (Appendix A.9). We speculate that this property of Lattice determines when it is to be preferred to uncorrelated noise, i.e., when there exists a low-dimensional solution to the task which allows the agent to take advantage of motor synergies. In environments where the dimensionality of the task is intrinsically low (*reach* and *object manipulation*), Lattice achieves strong performance. Further experiments in the *pose* and in the *object manipulation* tasks, performed with Lattice-SAC, confirm the result that Lattice-SAC is better than SAC for learning to manipulate objects (Appendix A.8).

The intrinsic dimensionality of the actions decreases as an effect of training (Appendix A.9). While training tends to reduce the dimensionality of the policy regardless of the action noise, the decrease is more marked with Lattice. The low dimensionality of the actions is explained by the action correlation matrices, which show increased cross-actuator coordination with Lattice (Fig. 6 B, D). We speculate that this is driven by the correlation structure of the noise. While the off-diagonal elements are close to 0 with uncorrelated noise, those of Lattice present a structure that resembles that of the action correlation matrix (Fig. 6 B, D), also consistent with the motivating example (Section 3). Indeed, if we consider the off-diagonal elements at position $(i, j)$, we have that $\text{Cov}(a_i, a_j) = W\text{Cov}(\mathbf{x})W^\top$, while $\mathbb{V}\left[\pi(\mathbf{a}|\mathbf{s})\right]_{i,j} = \alpha^2 \left(W\text{Diag}(S_{\mathbf{x}}^2\mathbf{x}^2)W^\top\right)_{i,j}$ (further details in Appendix A.5). In the case where $\alpha = 1$, $S_{\mathbf{x}}$ is the identity and the components of $\mathbf{x}$ are uncorrelated, then the two matrices are identical. While throughout the training the parameters of $S_{\mathbf{x}}$ can adapt, we can empirically see that the covariance matrix retains elements of its original structure.

## 7 Discussion and Limitations

We proposed Lattice, a method that leverages the synergies between action components encoded in the weights of the policy network to drive exploration based on the latent state. Latent exploration has proven effective in discovering high-reward and energy efficient policies in complex locomotion and muscle control tasks. Remarkably, Lattice outperforms random exploration in the *reach* and *object manipulation* tasks, with a large reduction in energy consumption. We showed that perturbing the last layer of the policy network introduces enough bias into the action noise. Importantly, this specific perturbation allowed us to find an analytical expression for the action distribution, which makes Lattice a straightforward enhancement for any on-policy or off-policy continuous control algorithm.

This sets Lattice apart from other forms of exploration using an auxiliary policy to collect experience, as they see their application limited to off-policy RL. Furthermore, knowing the action distribution is fundamental for those off-policy algorithms learning a stochastic policy, such as SAC. On the other hand, modeling the action distribution with a multivariate Gaussian with full covariance matrix comes at additional computational cost. In fact, Lattice introduces a training overhead over gSDE and PPO/SAC (approx. 20%-30%, depending on the hardware). This is due to the additional matrix multiplications required to estimate the action probabilities (Eq. 1). This limitation opens an interesting research direction, e.g., introducing sparsity constraints in the distribution matrices. For example, $S_{\mathbf{a}}$ and $S_{\mathbf{x}}$ could be forced to be diagonal or low-rank, without preventing motor synergies to be included into the noise distribution. We leave this investigation to future work.

Biological motor learning finds efficient and robust solutions [12, 13, 52–55]. Lattice also discovers low-energy solutions and it will be an interesting future question, if the brain is also performing some form of latent-driven exploration. Beyond this specific question, we are enthusiastic that advances in musculoskeletal simulators [18] as well as reinforcement learning are opening up exciting avenues for Neuroscience [56].

Our research aims to facilitate the training of motor control policies for complex tasks. Lattice might be employed as a tool in Artificial Intelligence, Robotics and Neuroscience, contributing to the creation of powerful autonomous agents. The energy efficiency of the learned policies might be of great interest for energy-sensitive robotics applications for reducing the carbon footprint of humans. While the advancement of the research in this field is a great opportunity, the related concerns should be carefully addressed [57].

## Acknowledgments

We are grateful to Adriana Rotondo, Stéphane D'Ascoli and other members of the Mathis Group for comments on earlier versions of this manuscript. Our work was funded by Swiss SNF grant (310030_212516) and EPFL. A.M.V.: Swiss Government Excellence Scholarship. A.Z.H.: EPFL School of Life Sciences Summer Research Program. We thank the Neuro-X institute at EPFL for supporting the travel expenses (A.S.C. and A.M.V).

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

# A   Appendix

## Contents

### A.1   Singe-joint arm: detailed calculations

**Case 1: action space noise.** By combining the equation of the angular acceleration $\ddot{\theta} = \alpha(a_e - a_f)$ with the distributions of the muscle activations $a_f \sim \mathcal{N}(\langle a_f \rangle, \sigma^2)$ and $a_e \sim \mathcal{N}(\langle a_e \rangle, \sigma^2)$, we can compute expected value and variance of the angular acceleration:

$$\mathbb{E}\left[\ddot{\theta}\right] = \mathbb{E}\left[\alpha(a_e - a_r)\right] = \alpha\left(\mathbb{E}\left[a_e\right] - \mathbb{E}\left[a_f\right]\right) = \alpha(\langle a_e \rangle - \langle a_r \rangle) \tag{3}$$

$$\mathbb{V}\left[\ddot{\theta}\right] = \mathbb{V}\left[\alpha(a_e - a_r)\right] = \alpha^2\left(\mathbb{V}\left[a_e\right] + \mathbb{V}\left[a_f\right]\right) = 2\alpha^2\sigma^2 \tag{4}$$

In the second derivation we have used the fact that $a_e$ and $a_r$ are independent random variables to compute their variances separately, and that $\mathbb{V}\left[-a_f\right] = \mathbb{V}\left[a_f\right]$.

**Case 2: latent space noise.** In this case we apply noise to the latent state, so that $\Delta\theta \sim \mathcal{N}(\langle\theta\rangle, \sigma^2)$. While it still holds that $a_f \sim \mathcal{N}(\langle a_f \rangle, \sigma^2)$ and $a_e \sim \mathcal{N}(\langle a_e \rangle, \sigma^2)$, the two random variables are no longer independent. This fact does not change the computation of the expectation:

$$\mathbb{E}\left[\ddot{\theta}\right] = \mathbb{E}\left[\alpha(a_e - a_r)\right] = \alpha\left(\mathbb{E}\left[a_e\right] - \mathbb{E}\left[a_f\right]\right) = \alpha(\langle a_e \rangle - \langle a_r \rangle) \tag{5}$$

For the variance, we need to make the dependence on $\theta$ explicit, using the formulas $a_e = 0.5 + \Delta\theta$ and $a_f = 0.5 - \Delta\theta$:

$$\mathbb{V}\left[\ddot{\theta}\right] = \mathbb{V}\left[\alpha(a_e - a_r)\right] = \mathbb{V}\left[2\alpha\Delta\theta\right] = 4\alpha^2\left(\mathbb{V}\left[\Delta\theta\right]\right) = 4\alpha^2\sigma^2 \tag{6}$$

We can observe that the variance is double in the case of latent space noise.

To test whether these results can be observed also in simulation, we considered the Elbow pose task of MyoSuite, where we trained a policy with PPO and Lattice exploration (with the hyperparameters specified in Table T4). The policy reaches a *solved* value of 0.95, meaning that the angle between the upper and the lower arm is in the immediate neighborhood of the target angle 95% of the simulation time. We assessed the effect of latent and action perturbations on the angular position by carrying out two simultaneous simulations of the environment per noise type. At every step, one simulation received a noisy action, while the other would receive a deterministic, noise-free action. We then registered the difference in elbow angle between the noisy and the deterministic simulation. To avoid having the results obfuscated by the cumulative perturbation over longer time periods, we synchronized the simulators after every step at the state achieved through the stochastic simulation.

To compute comparable results with latent and action noise, we had to make sure that the variance of each action component would be equivalent in both cases. We accomplished this by first running the simulation with latent noise, with the same variance for each latent state component. We then measured the variance that such perturbation would cause to each action component, and we used

this value to generate the action perturbation noise. This ensures that the difference in the variability of the elbow angle when applying latent or action noise is not due to a difference in scale, but only to the presence of off-diagonal elements in the noise covariance matrix.

## A.2 Lattice's Action Distribution Parameterization

In Lattice, the action vector $\mathbf{a}$ is a linear transformation of the perturbed latent vector $\tilde{\mathbf{x}} = \mathbf{x} + \boldsymbol{\epsilon}_{\mathbf{x}}$, further perturbed with independent action noise $\boldsymbol{\epsilon}_{\mathbf{a}}$:

$$\mathbf{a} = W\tilde{\mathbf{x}} + \boldsymbol{\epsilon}_{\mathbf{a}} = W(\mathbf{x} + \boldsymbol{\epsilon}_{\mathbf{x}}) + \boldsymbol{\epsilon}_{\mathbf{a}} \tag{7}$$

Furthermore, the latent noise $\boldsymbol{\epsilon}_{\mathbf{x}}$ and the action noise $\boldsymbol{\epsilon}_{\mathbf{a}}$ are defined as follows:

$$\boldsymbol{\epsilon}_{\mathbf{x}} = P_{\mathbf{x}}\mathbf{x} \quad \text{and} \quad \boldsymbol{\epsilon}_{\mathbf{a}} = P_{\mathbf{a}}\mathbf{x} \tag{8}$$

The elements of $P_{\mathbf{x}}$ and $P_{\mathbf{a}}$ are distributed as independent Gaussians:

$$(P_{\mathbf{x}})_{i,j} \sim \mathcal{N}\left(0, (S_{\mathbf{x}})_{i,j}^2\right) \quad \text{and} \quad (P_{\mathbf{a}})_{i,j} \sim \mathcal{N}\left(0, (S_{\mathbf{a}})_{i,j}^2\right) \tag{9}$$

where $S^{(\mathbf{x})}$ and $S^{(\mathbf{a})}$ are learnt parameter matrices whose elements represent the standard deviation of each element of the perturbation matrices. We can therefore compute the distribution of each element of the noise vectors. They are defined as:

$$(\boldsymbol{\epsilon}_{\mathbf{x}})_i = \alpha \sum_{j=1}^{N_{\mathbf{x}}} (P_{\mathbf{x}})_{i,j}\mathbf{x}_j \quad \text{and} \quad (\boldsymbol{\epsilon}_{\mathbf{a}})_i = \sum_{j=1}^{N_{\mathbf{x}}} (P_{\mathbf{a}})_{i,j}\mathbf{x}_j \tag{10}$$

meaning that they are the sum of independent Gaussian random variables. We can compute their mean and variance as follows:

$$\mathbb{E}\left[(\boldsymbol{\epsilon}_{\mathbf{x}})_i\right] = \mathbb{E}\left[\alpha \sum_{j=1}^{N_{\mathbf{x}}} (P_{\mathbf{x}})_{i,j}\mathbf{x}_j\right] = \alpha \sum_{j=1}^{N_{\mathbf{x}}} \mathbf{x}_j \mathbb{E}\left[(P_{\mathbf{x}})_{i,j}\right] = 0 \tag{11}$$

$$\mathbb{V}\left[(\boldsymbol{\epsilon}_{\mathbf{x}})_i\right] = \mathbb{V}\left[\alpha \sum_{j=1}^{N_{\mathbf{x}}} (P_{\mathbf{x}})_{i,j}\mathbf{x}_j\right] = \alpha^2 \sum_{j=1}^{N_{\mathbf{x}}} \mathbf{x}_j^2 \mathbb{V}\left[(P_{\mathbf{x}})_{i,j}\right] = \alpha^2 \sum_{j=1}^{N_{\mathbf{x}}} \mathbf{x}_j^2 (S_{\mathbf{x}})_{i,j}^2 \tag{12}$$

$$\mathbb{E}\left[(\boldsymbol{\epsilon}_{\mathbf{a}})_i\right] = \mathbb{E}\left[\sum_{j=1}^{N_{\mathbf{x}}} (P_{\mathbf{a}})_{i,j}\mathbf{x}_j\right] = \sum_{j=1}^{N_{\mathbf{x}}} \mathbf{x}_j \mathbb{E}\left[(P_{\mathbf{a}})_{i,j}\right] = 0 \tag{13}$$

$$\mathbb{V}\left[(\boldsymbol{\epsilon}_{\mathbf{a}})_i\right] = \mathbb{V}\left[\sum_{j=1}^{N_{\mathbf{x}}} (P_{\mathbf{a}})_{i,j}\mathbf{x}_j\right] = \sum_{j=1}^{N_{\mathbf{x}}} \mathbf{x}_j^2 \mathbb{V}\left[(P_{\mathbf{a}})_{i,j}\right] = \sum_{j=1}^{N_{\mathbf{x}}} \mathbf{x}_j^2 (S_{\mathbf{a}})_{i,j}^2 \tag{14}$$

The covariance of noise elements at different indices is $0$. Indeed, for $i \neq j$:

$$\text{Cov}\left[(\boldsymbol{\epsilon}_{\mathbf{x}})_i, (\boldsymbol{\epsilon}_{\mathbf{x}})_j\right] = \text{Cov}\left[\alpha \sum_{k=1}^{N_{\mathbf{x}}} (P_{\mathbf{x}})_{i,k}\mathbf{x}_k, \alpha \sum_{h=1}^{N_{\mathbf{x}}} (P_{\mathbf{x}})_{j,h}\mathbf{x}_h\right]$$
$$= \alpha^2 \sum_{k=1}^{N_{\mathbf{x}}} \sum_{h=1}^{N_{\mathbf{x}}} \mathbf{x}_k\mathbf{x}_h \text{Cov}\left[(P_{\mathbf{x}})_{i,k}, (P_{\mathbf{x}})_{j,h}\right] = 0 \tag{15}$$

$$\text{Cov}\left[(\boldsymbol{\epsilon}_{\mathbf{a}})_i, (\boldsymbol{\epsilon}_{\mathbf{a}})_j\right] = \text{Cov}\left[\sum_{k=1}^{N_{\mathbf{x}}} (P_{\mathbf{a}})_{i,k}\mathbf{x}_k, \sum_{h=1}^{N_{\mathbf{x}}} (P_{\mathbf{a}})_{j,h}\mathbf{x}_h\right]$$
$$= \sum_{k=1}^{N_{\mathbf{x}}} \sum_{h=1}^{N_{\mathbf{x}}} \mathbf{x}_k\mathbf{x}_h \text{Cov}\left[(P_{\mathbf{a}})_{i,k}, (P_{\mathbf{a}})_{j,h}\right] = 0 \tag{16}$$

where we used that different elements of $P_\mathbf{x}$ and $P_\mathbf{a}$ come from independent Gaussian distributions. Therefore, the joint distribution of the noise vectors $\epsilon_\mathbf{x}$ and $\epsilon_\mathbf{a}$ is Gaussian, with a diagonal covariance matrix:

$$\epsilon_\mathbf{x} \sim \mathcal{N}\left(\mathbf{0}, \alpha^2 \mathrm{Diag}\left(S_\mathbf{x}^2 \mathbf{x}^2\right)\right) \qquad \text{and} \qquad \epsilon_\mathbf{a} \sim \mathcal{N}\left(\mathbf{0}, \mathrm{Diag}\left(S_\mathbf{a}^2 \mathbf{x}^2\right)\right) \tag{17}$$

where the squares are to be intended element-wise. The distribution of $\mathbf{a}$ can be directly computed from the formula of the linear transformations of multivariate Gaussian distributions from Eq. (7):

$$\pi(\mathbf{a}|\mathbf{s}) = \mathcal{N}\left(W\mathbf{x}(\mathbf{s}), \mathrm{Diag}(S_\mathbf{a}^2 \mathbf{x}^2) + \alpha^2 W \mathrm{Diag}(S_\mathbf{x}^2 \mathbf{x}^2) W^\top\right) \tag{18}$$

where we have used the independence of $P_\mathbf{a}$ and $P_\mathbf{x}$ to find the covariance matrix of $\epsilon_\mathbf{a} + W \epsilon_\mathbf{x}$.

## A.3 Noise rescaling across networks of different size

Here, we will show that the variance of Lattice's generative noise model (Eq. (18)) depends the dimension of the latent state $x$. We thus, rescale $S_\mathbf{x}$ and $S_\mathbf{a}$ to be invariant to the nework size.

Derivation: We first observe that the standard deviation of each element of the noise vector $\epsilon_\mathbf{x}$ scales with the size of $\mathbf{x}$. Indeed:

$$\begin{aligned}
\mathbb{V}\left[(\epsilon_\mathbf{x})_i\right] &= \mathbb{V}\left[\alpha \sum_{j=1}^{N_\mathbf{x}} (P_\mathbf{x})_{i,j} \mathbf{x}_j\right] = \alpha^2 \sum_{j=1}^{N_\mathbf{x}} \mathbf{x}_j^2 \mathbb{V}\left[(P_\mathbf{x})_{i,j}\right] \\
&= \alpha^2 \langle \mathbb{V}\left[(P_\mathbf{x})_{i,j}\right]\rangle \sum_{j=1}^{N_\mathbf{x}} \mathbf{x}_j^2 = \alpha^2 N_\mathbf{x} \langle (S_\mathbf{x})_{i,j}^2\rangle \langle \mathbf{x}_j^2\rangle.
\end{aligned} \tag{19}$$

Therefore, if we consider the average value of $(S_\mathbf{x})_{i,j}^2$ and of $\mathbf{x}_j^2$ to be independent of the size of the latent state (at initialization time, it is the case, e.g., with the common Xavier initialization for the network parameters [58]), then we have that $\mathbb{V}\left[\epsilon_i\right]$ scales with $N_\mathbf{x}$. By applying the correction

$$\log S_\mathbf{x} = \log \tilde{S}_\mathbf{x} - 0.5 \log(N_\mathbf{x}) \tag{20}$$

we have that

$$(S_\mathbf{x})_{i,j}^2 = \exp\left(2(\log \tilde{S}_\mathbf{x} - 0.5 \log(N_\mathbf{x}))\right) = \frac{1}{N_\mathbf{x}} (\tilde{S}_\mathbf{x})_{i,j}^2 \tag{21}$$

In this way the initialization of $\log \tilde{S}_\mathbf{x}$ can be kept the same across networks with different latent state size. An identical argument is valid for $S_\mathbf{a}$, too.

## A.4 Conditions on the covariance matrix of the action distribution

First we show that the covariance matrix of the action distribution, defined as $W\Sigma_\mathbf{x} W^\top + \Sigma_\mathbf{a}$, is positive semidefinite by construction. We start by observing that the matrix $\Sigma_\mathbf{x} = \alpha^2 \mathrm{Diag}(S_\mathbf{x}^2 \mathbf{x}^2)$ and the matrix $\Sigma_\mathbf{a} = \mathrm{Diag}(S_\mathbf{a}^2 \mathbf{x}^2)$ are square diagonal matrices, whose elements are larger or equal than 0. We can therefore write $\Sigma_\mathbf{x} = \Sigma_\mathbf{x}^{\frac{1}{2}} \Sigma_\mathbf{x}^{\frac{1}{2}}$ and $\Sigma_\mathbf{a} = \Sigma_\mathbf{a}^{\frac{1}{2}} \Sigma_\mathbf{a}^{\frac{1}{2}}$, where the elevation to $\frac{1}{2}$ has to be applied to each element of the matrices. For any vector $\mathbf{y} \in \mathbb{R}^{N_\mathbf{x}}$, we have that

$$\mathbf{y}^\top \left(W\Sigma_\mathbf{x} W^\top + \Sigma_\mathbf{a}\right) \mathbf{y} = \mathbf{y}^\top \left(W\Sigma_\mathbf{x}^{\frac{1}{2}} \Sigma_\mathbf{x}^{\frac{1}{2}} W^\top + \Sigma_\mathbf{a}^{\frac{1}{2}} \Sigma_\mathbf{a}^{\frac{1}{2}}\right) \mathbf{y} = ||\Sigma_\mathbf{x}^{\frac{1}{2}} W^\top \mathbf{y}||^2 + ||\Sigma_\mathbf{a}^{\frac{1}{2}} \mathbf{y}||^2 \geq 0 \tag{22}$$

The regularization term we apply in Lattice consists in a multiple of the identity matrix by the coefficient $\gamma$, so that the minimum eigenvalue of the covariance matrix can never be lower than $\gamma$ itself.

Without a regularization term, the covariance matrix $W\Sigma_\mathbf{x} W^\top + \Sigma_\mathbf{a}$ might have 0 eigenvalues. In particular, this is common when using an activation function which promotes sparse latent representations, such as ReLU. In the limit case where $\mathbf{x}$ is the null vector, the covariance matrix is the null matrix.

Table T1: Task and reward parameters of Elbow pose, Finger pose, Finger reach and Hand pose.

| Task | Elbow pose Parameter | Value | Finger pose Parameter | Value | Finger reach Parameter | Value | Hand pose Parameter | Value |
|---|---|---|---|---|---|---|---|---|
| | Max steps | 100 | Max steps | 100 | Max steps | 100 | Max steps | 100 |
| | Pose threshold | 0.175 | Pose threshold | 0.35 | | | Pose threshold | 0.8 |
| | Target distance | 1 | Target distance | 1 | | | Target distance | 0.5 |
| Reward | Pose | 1 | Pose | 1 | Reach | 1 | Pose | 1 |
| | Bonus | 0 | Bonus | 0 | Bonus | 4 | Bonus | 0 |
| | Penalty | 1 | Penalty | 1 | Penalty | 50 | Penalty | 1 |
| | Action reg. | 0 | Action reg. | 0 | Action reg. | 0 | Action reg. | 0 |
| | Solved | 1 | Solved | 1 | Solved | 0 | Solved | 1 |
| | Done | 0 | Done | 0 | Done | 0 | Done | 0 |
| | Sparse | 0 | Sparse | 0 | Sparse | 0 | Sparse | 0 |

Table T2: Task and reward parameters of Hand reach, Baoding, Reorient and Pen.

| Task | Hand reach Parameter | Value | Baoding Parameter | Value | Reorient Parameter | Value | Pen Parameter | Value |
|---|---|---|---|---|---|---|---|---|
| | Max steps | 100 | Max steps | 200 | Max steps | 150 | Max steps | 100 |
| | | | Goal range x | (0.25, 0.25) | Goal pos. | (0, 0) | Goal orient. range | (-1, 1) |
| | | | Goal range y | (0.28, 0.28) | Goal rot. | (-0.785, 0.785) | | |
| Reward | Reach | 1 | Pos. dist. 1 | 1 | Pos.dist. | 1 | Pos. align | 0 |
| | Bonus | 4 | Pos. dist. 2 | 1 | Rot. dist. | 0.2 | Rot. align | 0 |
| | Penalty | 50 | Alive | 1 | Alive | 1 | Alive | 1 |
| | | | Action reg. | 0 | Action reg. | 0 | Action reg. | 0 |
| | | | Solved | 5 | Solved | 2 | Solved | 1 |
| | | | Done | 0 | Done | 0 | Done | 0 |
| | | | Sparse | 0 | Sparse | 0 | Sparse | 0 |
| | | | | | Pos. dist. diff. | 100 | Pos. align diff. | 100 |
| | | | | | Rot. dist. diff. | 10 | Rot. aligh diff. | 100 |

## A.5 Empirical covariance of the action

The covariance of the action components of a deterministic policy is given by:

$$
\begin{aligned}
\mathrm{Cov}(a_i, a_j) &= \mathrm{Cov}\left(\sum_k w_{i,k} x_k, \sum_h w_{j,h} x_h\right) \\
&= \sum_k \sum_h w_{i,k} w_{j,h} \mathrm{Cov}\left(x_k, x_h\right) \\
&= \left(W \mathrm{Cov}(\mathbf{x}) W^\top\right)_{i,j}
\end{aligned}
\tag{23}
$$

## A.6 Parameters of the PyBullet and MyoSuite environments

We used all the default parameters for the PyBullet environments [20], including the default reward function and episode length. The MyoSuite environments, instead, come with a defined metric for the success (the *solved* value), while the reward components are adjustable [18]. We keep the same reward components across all the algorithms, with values which consent achieving good *solved* values. In Table T1 and T2 we detail the environment and reward parameters we choose for our trainings.

We often add a new reward component, called *alive*, which is equal to 1 when the episode is not finished. It can be used to promote policies that do not trigger an early termination, e.g., by dropping the object. To be noted that we reduced the range of possible target poses in Hand pose, because no algorithm could solve the environment with the full range (without a curriculum), making the environment unsuitable to test the difference between standard and latent exploration.

## A.7 Hyperparameters of SAC, PPO, gSDE and Lattice

Our implementation was based on the library Stable Baselines 3 [45], and the code for Lattice will be shared in an open-source way.

Here, we summarize the parameters of SAC, gSDE-SAC and Lattice-SAC for the PyBullet locomotion tasks (Table T3) and the parameters of PPO, gSDE-PPO and Lattice-PPO in the MyoSuite muscle control tasks (Table T4 and T5). We also assessed the dependency of the final reward depending

Table T3: Parameters of SAC, gSDE-SAC and Lattice-SAC in the PyBullet locomotion tasks

| Task
Algorithms | Parameters | Ant | Half Cheetah | Walker | Hopper | Humanoid |
|---|---|---|---|---|---|---|
| SAC | Action normalization | Yes | Yes | Yes | Yes | Yes |
| | Buffer size | 300 000 | 300 000 | 300 000 | 300 000 | 300 000 |
| | Learning rate | 0.0003 | 0.0003 | 0.0003 | 0.0003 | 0.0003 |
| | Warmup steps | 10 000 | 10 000 | 10 000 | 10 000 | 10 000 |
| | Minibatch size | 256 | 256 | 256 | 256 | 256 |
| | Discount factor $\gamma$ | 0.98 | 0.98 | 0.98 | 0.98 | 0.98 |
| | Soft update coeff. $\tau$ | 0.02 | 0.02 | 0.02 | 0.02 | 0.02 |
| | Train frequency (steps) | 8 | 8 | 8 | 8 | 8 |
| | Num gradient steps | 8 | 8 | 8 | 8 | 8 |
| | Target update interval | 1 | 1 | 1 | 1 | 1 |
| | Entropy coefficient | auto | auto | auto | auto | auto |
| | Target entropy | auto | auto | auto | auto | auto |
| | Policy hiddens | [400, 300] | [400, 300] | [400, 300] | [400, 300] | [400, 300] |
| | Q hiddens | [400, 300] | [400, 300] | [400, 300] | [400, 300] | [400, 300] |
| | Activation | GELU | GELU | GELU | GELU | GELU |
| gSDE | Init log std | -3 | -3 | -3 | -3 | -3 |
| | Full std matrix | Yes | Yes | Yes | Yes | Yes |
| Lattice | Init log std | 0 | 1 | 1 | 1 | 0 |
| | Full std matrix | Yes | Yes | Yes | Yes | Yes |
| | Std clip | (0.001, 1) | (0.001, 10) | (0.001, 10) | (0.001, 10) | (0.001, 1) |
| | Std regularization | 0.001 | 0.001 | 0.001 | 0.001 | 0.001 |
| | $\alpha$ | 1 | 1 | 1 | 1 | 1 |

on the initial value of $\log \sigma$ in for Hand Pose and Reorient. The default value of 0 seems to perform optimally (Fig. F1).

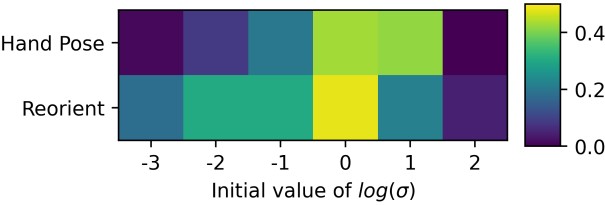

Figure F1: Final reward as a function of the initial value of the exploration standard deviation of Lattice, with period $T = 1$.

## A.8 Detailed reward and energy results of all experiments

We list the performance (energy and reward) of SAC, gSDE-SAC and Lattice-SAC in the PyBullet locomotion tasks (Table T6) and the performance of PPO, gSDE-PPO and Lattice-PPO in the MyoSuite tasks (Tables T7,T8, T9 and T10).

All results are averaged over 5 seeds and we report mean and standard error of mean.

## A.9 Evolution of the dimensionality of the policy during training

In the main text we analyzed how Lattice explores the environment (Section 6). As discussed in the main text, here we present the results across learning.

We studied how the dimensionality of the policy changes during training when the policy is trained with Lattice, compared to standard SAC and PPO. While the number of relevant components is similar at the beginning of the training, the policies trained with Lattice converge to actions that lie

Table T4: Parameters of PPO, gSDE-PPO and Lattice-PPO in Elbow pose, Finger pose, Hand pose and Finger reach.

| Task Algorithms | Parameters | Elbow pose | Finger pose | Hand pose | Finger reach |
|---|---|---|---|---|---|
| PPO | Action normalization | Yes | Yes | Yes | Yes |
| | Learning rate | 0.000025 | 0.000025 | 0.000025 | 0.000025 |
| | Batch size | 32 | 32 | 32 | 32 |
| | Gradient steps | 128 | 128 | 128 | 128 |
| | Num epochs | 10 | 10 | 10 | 10 |
| | Discount factor $\gamma$ | 0.99 | 0.99 | 0.99 | 0.99 |
| | Entropy coefficient | 0.0000036 | 0.0000036 | 0.0000036 | 0.0000036 |
| | Value function coefficient | 0.84 | 0.84 | 0.84 | 0.84 |
| | GAE $\lambda$ | 0.9 | 0.9 | 0.9 | 0.9 |
| | Clip parameter | 0.3 | 0.3 | 0.3 | 0.3 |
| | Max gradient norm | 0.7 | 0.7 | 0.7 | 0.7 |
| | Policy hiddens | [256, 256] | [256, 256] | [256, 256] | [256, 256] |
| | Critic hiddens | [256, 256] | [256, 256] | [256, 256] | [256, 256] |
| | Policy LSTM hiddens | 256 | 256 | 256 | 256 |
| | Critic LSTM hiddens | 256 | 256 | 256 | 256 |
| | Activation | ReLU | ReLU | ReLU | ReLU |
| gSDE | Init log std | -2 | -2 | -2 | -2 |
| | Full std matrix | No | No | No | No |
| Lattice | Init log std | 1 | 0 | 1 | 0 |
| | Full std matrix | No | No | No | No |
| | Std clip | (0.001, 10) | (0.001, 10) | (0.001, 10) | (0.001, 10) |
| | Std regularization | 0 | 0 | 0 | 0 |
| | $\alpha$ | 1 | 1 | 1 | 1 |

on a lower dimensional manifold (Fig. F2). We hypothesize that this is the main reason why Lattice leads to more energy-efficient policies. Indeed, the energy consumption is similar for both Lattice and SAC/PPO at initialization and diverges during training with Lattice achieving more energy-efficient policies; this result is also consistent across seeds(Fig. F3). Overall, the final policies trained with lattice have lower dimensionality, consistently across environments and random seeds (Fig. F4).

Table T5: Parameters of PPO, gSDE-PPO and Lattice-PPO in Hand reach, Baoding, Reorient and Pen.

| Task Algorithms | Parameters | Hand reach | Baoding | Reorient | Pen |
|---|---|---|---|---|---|
| PPO | Action normalization | Yes | Yes | Yes | Yes |
| | Learning rate | 0.000025 | 0.000025 | 0.000025 | 0.000025 |
| | Batch size | 32 | 32 | 32 | 32 |
| | Gradient steps | 128 | 128 | 128 | 128 |
| | Num epochs | 10 | 10 | 10 | 10 |
| | Discount factor $\gamma$ | 0.99 | 0.99 | 0.99 | 0.99 |
| | Entropy coefficient | 0.0000036 | 0.0000036 | 0.0000036 | 0.0000036 |
| | Value function coefficient | 0.84 | 0.84 | 0.84 | 0.84 |
| | GAE $\lambda$ | 0.9 | 0.9 | 0.9 | 0.9 |
| | Clip parameter | 0.3 | 0.3 | 0.3 | 0.3 |
| | Max gradient norm | 0.7 | 0.7 | 0.7 | 0.7 |
| | Policy hiddens | [256, 256] | [256, 256] | [256, 256] | [256, 256] |
| | Critic hiddens | [256, 256] | [256, 256] | [256, 256] | [256, 256] |
| | Policy LSTM hiddens | 256 | 256 | 256 | 256 |
| | Critic LSTM hiddens | 256 | 256 | 256 | 256 |
| | Activation | ReLU | ReLU | ReLU | ReLU |
| gSDE | Init log std | -2 | -2 | -2 | -2 |
| | Full std matrix | No | No | No | No |
| Lattice | Init log std | 0 | 0 | 0 | 0 |
| | Full std matrix | No | No | No | No |
| | Std clip | (0.001, 10) | (0.001, 10) | (0.001, 10) | (0.001, 10) |
| | Std regularization | 0 | 0 | 0 | 0 |
| | $\alpha$ | 1 | 1 | 1 | 1 |

Table T6: Detailed results in the PyBullet locomotion environments. Results are averaged over N=5 seeds.

| | Ant Energy | Reward | Hopper Energy | Reward | Walker Energy | Reward | Half cheetah Energy | Reward | Humanoid Energy | Reward |
|---|---|---|---|---|---|---|---|---|---|---|
| SAC | 0.23 ± 0.01 | 3381 ± 30 | 0.26 ± 0.01 | 2417 ± 106 | 0.27 ± 0.0 | 2741 ± 81 | 0.23 ± 0.0 | 2934 ± 27 | 0.12 ± 0.0 | 2122 ± 169 |
| SAC-gSDE period 4 | 0.23 ± 0.0 | 3978 ± 14 | 0.23 ± 0.01 | 2356 ± 17 | 0.26 ± 0.0 | 2728 ± 74 | 0.23 ± 0.0 | 3191 ± 125 | 0.12 ± 0.0 | 2114 ± 126 |
| SAC-gSDE period 8 | 0.23 ± 0.0 | 3962 ± 7 | 0.25 ± 0.01 | 2234 ± 86 | 0.25 ± 0.0 | 2746 ± 39 | 0.28 ± 0.01 | 2752 ± 23 | 0.12 ± 0.0 | 1927 ± 120 |
| SAC-gSDE episode | 0.24 ± 0.01 | 3796 ± 48 | 0.24 ± 0.01 | 2472 ± 64 | 0.26 ± 0.0 | 2822 ± 32 | 0.24 ± 0.0 | 3081 ± 93 | 0.11 ± 0.0 | 2460 ± 160 |
| SAC-Lattice (ours) | 0.25 ± 0.0 | 3544 ± 212 | 0.23 ± 0.01 | 2610 ± 78 | 0.29 ± 0.0 | 2718 ± 92 | 0.27 ± 0.01 | 2900 ± 67 | 0.11 ± 0.0 | 2742 ± 77 |
| SAC-Lattice period 4 (ours) | 0.25 ± 0.0 | 3926 ± 78 | 0.23 ± 0.01 | 2446 ± 133 | 0.29 ± 0.0 | 2541 ± 129 | 0.26 ± 0.0 | 2964 ± 44 | 0.11 ± 0.0 | 2798 ± 197 |
| SAC-Lattice period 8 (ours) | 0.24 ± 0.0 | 3686 ± 100 | 0.24 ± 0.01 | 2621 ± 86 | 0.3 ± 0.0 | 2641 ± 121 | 0.26 ± 0.0 | 3031 ± 23 | 0.11 ± 0.0 | 2358 ± 77 |
| SAC-Lattice episode (ours) | 0.25 ± 0.0 | 3845 ± 86 | 0.22 ± 0.01 | 2586 ± 23 | 0.28 ± 0.0 | 2661 ± 130 | 0.25 ± 0.0 | 2948 ± 24 | 0.11 ± 0.0 | 2901 ± 211 |

Table T7: Detailed results in MyoSuite environments: Elbow Pose and Finger Pose. Results are averaged over N=5 seeds.

| | Elbow pose Energy | Reward | Solved | Finger pose Energy | Reward | Solved |
|---|---|---|---|---|---|---|
| PPO | 0.21 ± 0.01 | 88.29 ± 0.33 | 0.95 ± 0.0 | 0.05 ± 0.04 | 81.61 ± 3.08 | 0.96 ± 0.01 |
| PPO-gSDE period 4 | 0.18 ± 0.0 | 71.15 ± 3.84 | 0.84 ± 0.03 | 0.02 ± 0.01 | 47.68 ± 11.88 | 0.78 ± 0.07 |
| PPO-Lattice (ours) | 0.19 ± 0.01 | 85.55 ± 0.67 | 0.94 ± 0.0 | 0.04 ± 0.0 | 74.35 ± 4.51 | 0.91 ± 0.03 |
| PPO-Lattice period 4 (ours) | 0.27 ± 0.01 | 62.57 ± 1.35 | 0.77 ± 0.01 | 0.05 ± 0.02 | 52.57 ± 9.0 | 0.78 ± 0.05 |

Table T8: Detailed results in MyoSuite environments: Finger reach and Hand pose. Results are averaged over N=5 seeds.

| | Finger reach Energy | Reward | Solved | Hand pose Energy | Reward | Solved |
|---|---|---|---|---|---|---|
| PPO | 0.2 ± 0.02 | 242.72 ± 15.53 | 0.2 ± 0.02 | 0.04 ± 0.0 | -25.85 ± 3.05 | 0.54 ± 0.03 |
| PPO-gSDE period 4 | 0.07 ± 0.02 | 257.97 ± 23.81 | 0.22 ± 0.03 | 0.04 ± 0.0 | -74.13 ± 7.81 | 0.24 ± 0.05 |
| PPO-Lattice (ours) | 0.04 ± 0.01 | 337.07 ± 15.62 | 0.33 ± 0.02 | 0.04 ± 0.0 | -44.8 ± 8.75 | 0.42 ± 0.06 |
| PPO-Lattice period 4 (ours) | 0.05 ± 0.0 | 206.46 ± 52.45 | 0.18 ± 0.05 | 0.03 ± 0.0 | -99.59 ± 9.48 | 0.11 ± 0.04 |

Table T9: Detailed results in MyoSuite environments: Hand reach and Baoding. Results are averaged over N=5 seeds.

| | Hand reach | | | Baoding | | |
| --- | --- | --- | --- | --- | --- | --- |
| | Energy | Reward | Solved | Energy | Reward | Solved |
| PPO | $0.09 \pm 0.0$ | $581.4 \pm 30.37$ | $0.52 \pm 0.08$ | $0.08 \pm 0.0$ | $573.46 \pm 73.24$ | $0.4 \pm 0.07$ |
| PPO-gSDE period 4 | $0.07 \pm 0.0$ | $462.72 \pm 25.97$ | $0.21 \pm 0.06$ | $0.07 \pm 0.01$ | $437.12 \pm 91.91$ | $0.26 \pm 0.09$ |
| PPO-Lattice (ours) | $0.04 \pm 0.0$ | $586.9 \pm 19.19$ | $0.54 \pm 0.05$ | $0.04 \pm 0.0$ | $627.31 \pm 69.01$ | $0.44 \pm 0.07$ |
| PPO-Lattice period 4 (ours) | $0.07 \pm 0.0$ | $556.1 \pm 11.14$ | $0.44 \pm 0.03$ | $0.07 \pm 0.01$ | $471.16 \pm 99.66$ | $0.29 \pm 0.1$ |

Table T10: Detailed results in MyoSuite environments: Reorient and Pen. Results are averaged over N=5 seeds.

| | Reorient | | | Pen | | |
| --- | --- | --- | --- | --- | --- | --- |
| | Energy | Reward | Solved | Energy | Reward | Solved |
| PPO | $0.04 \pm 0.01$ | $233.74 \pm 16.83$ | $0.33 \pm 0.05$ | $0.07 \pm 0.01$ | $140.83 \pm 9.77$ | $0.16 \pm 0.03$ |
| PPO-gSDE period 4 | $0.04 \pm 0.01$ | $187.99 \pm 27.04$ | $0.2 \pm 0.08$ | $0.07 \pm 0.0$ | $193.85 \pm 13.42$ | $0.42 \pm 0.06$ |
| PPO-Lattice (ours) | $0.01 \pm 0.0$ | $283.07 \pm 14.99$ | $0.48 \pm 0.05$ | $0.04 \pm 0.0$ | $169.72 \pm 15.98$ | $0.29 \pm 0.06$ |
| PPO-Lattice period 4 (ours) | $0.03 \pm 0.01$ | $250.13 \pm 13.6$ | $0.38 \pm 0.04$ | $0.06 \pm 0.0$ | $191.8 \pm 15.75$ | $0.39 \pm 0.08$ |

Table T11: SAC trained on a subset of the MyoSuite environments: Finger Pose, Hand Pose, Reorient and Pen. Results are averaged over N=3 seeds.

| | Finger pose SAC | | Hand pose SAC | | Reorient SAC | | Pen SAC | |
| --- | --- | --- | --- | --- | --- | --- | --- | --- |
| | Energy | Solved | Energy | Solved | Energy | Solved | Energy | Solved |
| SAC | $0.04 \pm 0.0$ | $0.96 \pm 0.0$ | $0.04 \pm 0.0$ | $0.84 \pm 0.03$ | $0.06 \pm 0.0$ | $0.55 \pm 0.03$ | $0.07 \pm 0.0$ | $0.39 \pm 0.14$ |
| SAC-gSDE episode | $0.04 \pm 0.0$ | $0.96 \pm 0.0$ | $0.04 \pm 0.0$ | $0.84 \pm 0.0$ | $0.07 \pm 0.0$ | $0.5 \pm 0.03$ | $0.07 \pm 0.0$ | $0.17 \pm 0.02$ |
| SAC-Lattice episode (ours) | $0.02 \pm 0.01$ | $0.92 \pm 0.01$ | $0.04 \pm 0.0$ | $0.73 \pm 0.02$ | $0.06 \pm 0.0$ | $0.67 \pm 0.03$ | $0.08 \pm 0.0$ | $0.59 \pm 0.03$ |

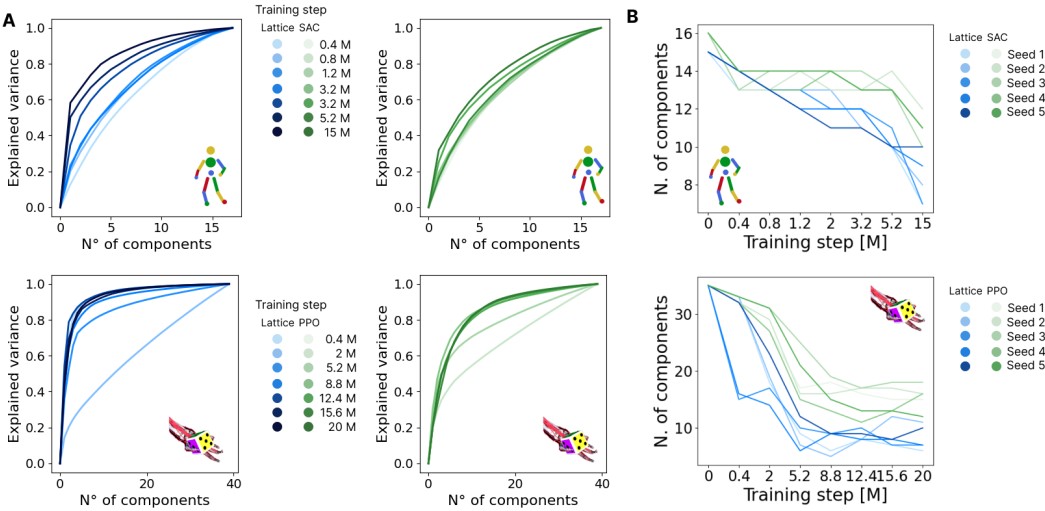

Figure F2: **A** Both in the Humanoid (top) and in the Reorient (bottom) motor control tasks, policies trained with Lattice require fewer principal components to explain the variance of the actions. The graphs, generated by testing policies at different stages of the training, highlight how the number of components decreases almost uniformly throughout the training, both with Lattice and with independent action noise. However, the effect is much stronger with Lattice. Result for one seed. **B** Number of principal components that explain at least 90% of the variance with respect to the training step for different seeds (N=5). For both Humanoid (top) and in the Reorient (bottom) tasks, Lattice reaches lower principal components during training compared to SAC/PPO.

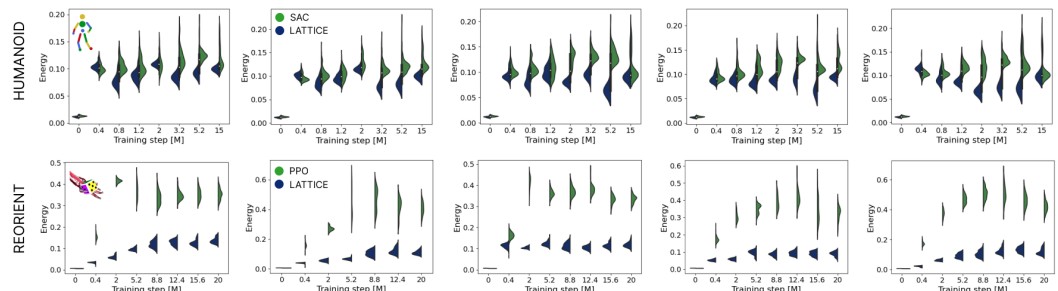

Figure F3: Distribution of the energy consumption across training episodes for different seeds (column) in the Humanoid (top) and Reorient (bottom) task. For both, the energy consumption is similar at the beginning and (relatively quickly diverges during training. Once the policy is trained, LATTICE shows a more energy-efficient policy.

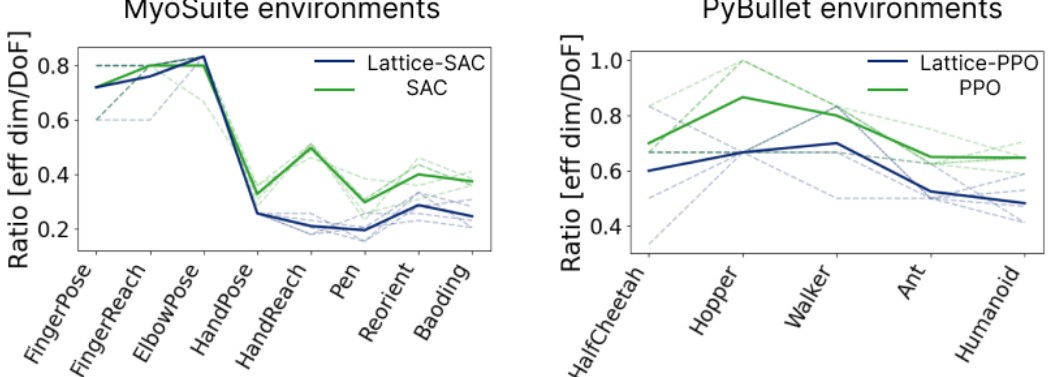

Figure F4: Analysis of the effective dimensionality of the policies obtained with Lattice-SAC and Lattice-PPO versus standard SAC and PPO in the PyBullet (left) and Myosuite (right) environments. The dimensionality is estimated by running 100 test episodes per environment (5 random seeds), computing the principal component analysis of the actions counting how many principal components are necessary to reach 90% cumulative explained variance, rescaled by the number of action components. We can see that, consistently across most environments and random seeds (dotted lines in the plot), Lattice leads to lower dimensional policies.

