# OpenReview forum: "Latent exploration for Reinforcement Learning"
_NeurIPS.cc/2023/Conference — NeurIPS 2023 poster_

### Official Review · Reviewer_dZAS · 2023-07-04

**Soundness:** 3 good
**Presentation:** 2 fair
**Contribution:** 3 good
**Rating:** 7
**Confidence:** 4

**Summary:**

This paper explores the exploration problem in RL algorithms by considering the action-space correlation. The approach is motivated by the fact that over-actuated systems with high-dof action space require correlated control actions. Such action-space correlation is ignored by action-space random perturbation in a standard RL algorithm which results in inefficient exploration. This work proposes to add random perturbation in the latent space in the last layer of the policy network to impose the proper action structure in the perturbation distribution. The presented approach can be potentially integrated into most existing RL algorithms to enhance the exploration capability and is shown to perform great, especially in high-dim musculoskeletal environments.

**Strengths:**

1. The motivation of the work is clear and intuitive. The provided toy example is a very illustrative example to motivate the idea of using correlated action perturbation.

2. The proposed method is concise and seems to be universal in that it can be easily integrated into most RL frameworks to improve exploration efficiency.

3. The experimental results are convincing to show the effectiveness of the proposed Lattice method.

**Weaknesses:**

1. The selection of the initial logstd for the $S$ matrices seems to be not straightforward and might need extra experiments to determine.

2. The presented Lattice method relies on a good learned policy to impose the action correlation structure, which might result in local-optimal policy (also see point 3 in the next section).

3. The exposition of the paper can be further improved to be more clear and consistent.

   a) The second motivating example (the one in simulation) is confusing.

   b) In Line 148, $N_l$ is used for the latent space size, while in Figure 1, $N_x$ is used for latent space size.

   c) Line 168: $\epsilon_a = P_a a \Rightarrow \epsilon_a = P_a x$

   d) It was never mentioned in the main paper about the separate action space perturbation until $\Sigma_a$ appears in Line 152.

**Questions:**

1. The two images in Figure 2(B) is hard to parse. Are those images obtained from the action perturbation training or the latent perturbation approach?

2. In the MyoSuite environments (Figure 4), I noticed Lattice-PPO (T=1) consistently works better than Lattice-PPO (T=4) except for the **Pen** task. Does it mean time-space correlation is not important in general? Do you have any idea on why T=1 works better than T=4 and what is special in **Pen** task that makes T=4 preferable?

3. The proposed Lattice approach benefits the action structure learned by the policy and imposes this learned structure to the action perturbation. However, on the other side is this also a limitation of this approach? Will Lattice suppress the emergence of novel control modes and result in a local-optimal policy? If this is not a problem, which part of the algorithm is the key to addressing this issue?

4. It would be valuable to also ablate the $P_ax$ term in the method. Is this the key to addressing the last point?

5. gSDE (T=4) seems to work consistently worse than other approaches (even worse than vanilla PPO) in all MyoSuite environments. It would be interesting to see the performance of gSDE-PPO (T=1) which differs from PPO on whether having state-dependent perturbation. If gSDE-PPO (T=1) does not work well either, then it might not be reasonable to have $\alpha$ to toggle between gSDE and Lattice. Instead, using $\alpha$ to toggle between regular RL perturbation and latent space perturbation might work better?

6. Currently SAC and its variants are used for the Pybullet environments while PPO and its variants are used for MyoSuite environments. It would be also great to provide all of those algorithms on all benchmark problems (maybe in Appendix).

**Limitations:**

As mentioned in point 3 in the **Questions** section.

---

> ### Author Rebuttal · Authors · 2023-08-09
>
> We thank the reviewer for their comments and positive evaluation.
>
> *Selection of log_std*
>
> We have run additional ablation experiments on Hand Pose and Reorient, where we have tested initial values for log_std between -3 and 2. The following results show that values around 0 are optimal:
>
> |std_init |-3|-2|-1|0|1|2|
> |:-|:-|:-|:-|:-|:-|:-|
> |Reorient|0.1815 ± 0.0411|0.3058 ± 0.0624|0.3051 ± 0.0581|0.4824 ± 0.0468|0.2179 ± 0.0808|0.0442 ± 0.0232|
> |Hand Pose|0.0109 ± 0.0094|0.0841 ± 0.0728|0.2006 ± 0.0726|0.4303 ± 0.0528|0.4184 ± 0.0638|0.0001 ± 0.0001|
>
> Thanks to the rescaling described in Appendix A3, these values should perform well with different hidden layer sizes.
>
> *Clarification of Figure 2*
>
> We modified the explanation for improved clarity:
> "To corroborate this finding about the properties of latent versus action perturbation, we considered a policy network trained to control a realistic musculoskeletal arm model, implemented in MyoSuite [17], featuring three flexor muscles and three extensor muscles. The policy can learn to accurately reach a target angle in every episode, and while doing so learns to alternately activate the extensor or the flexor muscle group (Fig. 2B, left). The training procedure is detailed in Appendix A1. We collected a dataset of 100 episodes where the latent state of the policy network was perturbed with gaussian noise. While applying independent random noise to the muscle activations produces a diagonal covariance matrix, perturbing the latent state of the network leads to a full covariance matrix. As hypothesized, the last layer of the policy network indeed learned to distinguish between agonist and antagonist muscles, namely, the covariance shows positive correlation among muscles of the same group (Fig. 2B, right). We then tested whether this covariance structure leads to higher variance in the task space (joint angle). We collected 100 episodes where, at each step, we computed an action with action space perturbation and one action with latent state perturbation. For consistency of evaluation, the perturbation in the action space was sampled with the same variance magnitude induced by the latent perturbation. We executed each pair of actions in two cloned environments and compared the variance of the joint angles. The latent state perturbation introduces higher variance in the joint angles (Fig. 2C), thus driving more diverse kinematics and thus wider exploration."
>
> The action correlation matrix (heatmap on the left) is relative to a policy network trained with Lattice. We will further clarify the training procedure in Appendix A1.
>
> *Remarks b) - d)*
>
> In the updated manuscript, we fixed the typos and we specified that we also consider an independent action noise components with diagonal covariance Sigma_a.
>
> *Time correlation in the MyoSuite experiments*
>
> As previously found in [1], time correlation paired with PPO is less effective than when it’s paired with SAC. Our experiments confirm this general trend for muscle-skeletal models. The Pen environment is a notable exception. We speculate that this might be related to the active stabilisation required to balance the pen, which is usually held between two or three fingers. In the other object manipulation environments (Baoding and Reorient), gravity is countered by keeping the palm below the object. Time-correlated noise might help avoiding the pen to fall, preventing an early termination of the episode.
>
> *Ablation of $P_a x$*
>
> Latent noise alone might not be enough to learn good policies, as it might excessively bias the noise with correlation across action components which might not be ideal, especially in the early stages of the training. We ran an ablation, where we suppress the Pax term of the state-dependent noise. We limited this analysis to episodic updates of the perturbation matrix in the PyBullet environments, because this experiment can be performed in about 2 days (5 random seeds per environment). We report the average reward ± standard error:
> | |Ant|Hopper|Walker|Half Cheetah|Humanoid|
> |:-|:-|:-|:-|:-|:-|
> |$P_a x$|**3846 ± 87**|**2587 ± 24**|**2662 ± 130**|**2948 ± 25**|2902 ± 212|
> |No $P_a x$|938 ± 70|139 ± 91|365 ± 58|-904 ± 337|2960 ± 129|
>
> Unsurprisingly, the performance is very low in Ant, Hopper, Walker and Half Cheetah, showing that independent action noise is necessary for the correct exploration of these environments. Maybe surprisingly, the performance is excellent in the Humanoid, where the perturbation of many degrees of freedom might contribute to provide enough variability to avoid early convergence to a local minimum.
>
> *Performance of gSDE period 1 in MyoSuite: correlated vs. state-dep noise*
>
> We considered the three pose environments of MyoSuite, as Lattice was on par or a little worse than standard exploration, to verify that the lower performance was due to the introduction of correlated noise, rather than state-dependent exploration. We ran gSDE with period=1 (no time correlation, but state-dependent noise) in Elbow Pose, Finger Pose and Hand Pose (3 seeds per environment). We report the results in terms of solved fraction ± standard error:
>
> | |Elbow Pose|Finger Pose|Hand Pose|
> |:-|:-|:-|:-|
> |gSDE period 1| 0.9014 ± 0.0219|**0.9843 ± 0.0016**|**0.5248 ± 0.0475**|
> |Lattice period 1|**0.9385 ± 0.0043**|0.9137 ± 0.0267|0.4184 ± 0.0638|
>
> These results support the fact that toggling between gSDE and Lattice can be useful also when the period is 1. In fact, alpha=0 improves Finger pose and Hand pose, bridging the gap with white noise.
>
> *Additional experiments: SAC in MyoSuite and PPO in PyBullet*
>
> We propose to run at least a subset of the proposed experiments for the camera-ready version, as we do not have enough computational resources to satisfy this request in the rebuttal period.
>
> [1] Antonin Raffin, Jens Kober, and Freek Stulp. "Smooth exploration for robotic reinforcement learning." CORL 2022.

---

> > ### Comment · Reviewer_dZAS · 2023-08-12
> >
> > Thank the authors for their detailed responses and they have addressed most of my concerns if the additional promised experiments can be conducted in the camera-ready version. I reaffirm my acceptance recommendation.

---

### Official Review · Reviewer_Q9SL · 2023-07-04

**Soundness:** 4 excellent
**Presentation:** 4 excellent
**Contribution:** 3 good
**Rating:** 6
**Confidence:** 4

**Summary:**

This paper proposes an exploration method that adds noise to the last linear layer (latent) of the policy network to incorporate time-dependent noise. The method is tested on locomotion tasks and evaluated on the metrics of performance and energy consumption. Evaluation is done on PPO and SAC base algorithm and an existing time-correlated baseline, gSDE.

**Strengths:**

- Tackles an important problem of exploration in tasks with continuous action.
- The proposed method is rather simple and presented well.
- Thorough experiments with necessary details and hyperparameters are presented.
- Performance is evaluated on return and energy conservation which is important for practical implications.


**Weaknesses:**

- Unclear why the proposed method enabled energy-saving policy compared to another time-correlated baseline.
- Energy comparison with gSDE is missing, which makes it difficult to evaluate how the noise injected in the latent layer is helping.
- Few cases, the proposed method outperforms the existing method in return performance.


**Questions:**

It is unclear why the proposed method enabled energy-saving policies compared to another time-correlated baseline. The comparison is conducted between PPO/SAC and not with gSDE (time-correlated). It is unclear how much the injected noise in the latent space contributes compared to the time-correlated component.

In Figure 4, the results are mixed. It is important to understand why time correlation sometimes worsens performance. It might be interesting to consider whether the proposed task requires independent or time-correlated noise.

The description of Figure 6 is currently difficult to understand. Simplifying the description would make it easier to comprehend the implications of the energy comparison.

The main contribution of the proposed method lies in the addition of noise to the latent space. Therefore, it is crucial to comprehend how this component is implemented. In terms of empirical return, the performance of the proposed method is mostly similar to the baseline. However, it is important to investigate how the introduction of noise in the latent space contributes to the observed energy consumption improvements. This information would provide a better understanding of the implications of the proposed method.


**Limitations:**

Yes.

---

> ### Author Rebuttal · Authors · 2023-08-08
>
> We thank the reviewer for their comments and strong evaluation (soundess, presentation and contribution) and excellent questions. We hope to be able to provide sufficient motivation to reconsider the assigned score.
>
> *It is unclear why the proposed method enabled energy-saving policies compared to another time-correlated baseline. The comparison is conducted between PPO/SAC and not with gSDE (time-correlated). It is unclear how much the injected noise in the latent space contributes compared to the time-correlated component.*
>
> We would like to remark that the comparison presented in figure 5 represents Lattice-PPO without time correlation (namely, with the exploration matrices resampled at every step). We made this choice because the time correlation might also have an impact on the energy consumption (as the reviewer suggests), and we aimed to isolate the effect of the action correlation introduced by Lattice. Furthermore, time correlation has proven quite ineffective in combination with PPO (as it was previously found in [1]). Therefore, because of the often low performance of time-correlated exploration in these locomotion environments, we believe the most meaningful comparison is between vanilla PPO and Lattice-PPO with no time correlation and we included it in the main section of the manuscript. The details about the performance and energy consumption of all policies are included in table T6-T10 of the appendix.
> In short, in the Humanoid environment and in the MyoSuite tasks, the best performing policies using Lattice are always more efficient than the best policies using white noise or gSDE exploration. With gSDE, when time correlation improves energy efficiency, it does so at the cost of a large fraction of the reward. For example, gSDE period 4 finds energy-efficient policies in Elbow Pose and Hand Pose, but the reward is much lower than without time correlation. Importantly, Lattice can find the most energy-efficient policy across many environments without compromising the reward (Humanoid, Finger Reach, Baoding, Hand Reach, Reorient, Pen)
>
> *In Figure 4, the results are mixed. It is important to understand why time correlation sometimes worsens performance. It might be interesting to consider whether the proposed task requires independent or time-correlated noise.*
>
> We empirically find that PPO does not benefit from time-correlated exploration as much as SAC, confirming the findings of [1]. We believe that this is at least in part due to the on-policy nature of PPO. Previous results on time-correlated noise (gSDE [1], Ornstein-Uhlenbeck noise [2] and pink noise [3]) focus on off-policy RL algorithms. Lattice, instead, in its non-time-correlated version (period 1), is an effective addition also to on-policy PPO, as showcased in the MyoSuite environments.
>
> *The description of Figure 6 is currently difficult to understand. Simplifying the description would make it easier to comprehend the implications of the energy comparison.*
>
> We have rephrased and extended the caption in the following way:
> “Figure 6: A Left: Graph of the fraction of noise allocated to each group of action components by a stochastic policy trained in the Humanoid environment with SAC and Lattice-SAC. In the Humanoid, Lattice tends to focus most of the variance on the task-relevant actuators (legs). Middle: Distribution of the energy consumption. Each bar represents the number of test episodes falling in the corresponding energy consumption interval. Right: Cumulative explained variance of the actions’ principal components, computed from a dataset of 100 test episodes per seed. Shaded area represents 95% confidence interval across training seeds. More principal components are required to explain the same fraction of variance in SAC and PPO versus Lattice-SAC and Lattice-PPO. B Heatmap of the correlation matrix for the action space (left column) and noise of the action (right column) for SAC (top row) and Lattice-SAC (bottom row). C-D Same as A-B but analyzing a policy trained on the MyoSuite Reorient task with PPO and Lattice-PPO. Lattice tends to induce sharper correlation between action components. Local patterns of correlation between action components can be recognized in the noise covariance matrix.
> “
>
> *The main contribution of the proposed method lies in the addition of noise to the latent space. Therefore, it is crucial to comprehend how this component is implemented. (...)*
>
> We hope our previous answers clarified the differences between the policies trained with Lattice versus uncorrelated action noise and how they influence energy consumption. We remark that these effects are not a consequence of the time correlation, but rather of the action component correlation, making them a distinguishing property of Lattice. The motivating example provides a high-level intuition as to why Lattice might cause an energy saving, while improving exploration. In highly redundant motor systems, different policies can produce the same observed behavior. For example, in a flexor-extensor system, the angular acceleration is (in first approximation) proportional to the difference between the activation of the two muscles. If both muscles are activated together, the resulting acceleration is 0, but the body spends energy to maintain this active equilibrium. When the environment is explored by randomly activating all muscles independently, it is more likely to obtain policies with non-zero base activation of the muscles. If the muscles activations are perturbed according to their synergy, it will be more likely to find policies with low base activation of the muscle, improving efficiency.
>
> [1] Raffin, Antonin, et al. "Smooth exploration for robotic reinforcement learning." CORL 2022.
>
> [2] Lillicrap, Timothy P., et al. "Continuous control with deep reinforcement learning." arXiv 2015.
>
> [3] Eberhard, Onno, et al. "Pink noise is all you need: Colored noise exploration in deep reinforcement learning." ICLR 2023.

---

> > ### Comment · Reviewer_Q9SL · 2023-08-15
> >
> > I would like to thank the authors for their detailed responses. Most of my concerns have been addressed. However, the empirical performance is still comparable to the baselines, and in some cases, it even worsens the performance (as seen in Figure 4). Investigating this aspect could enhance the quality of the paper. As a result, I have raised my rating from 4 to 6.

---

### Official Review · Reviewer_p1VW · 2023-07-21

**Soundness:** 3 good
**Presentation:** 3 good
**Contribution:** 3 good
**Rating:** 5
**Confidence:** 4

**Summary:**

This paper proposes a new method to conduct action exploration based on the latent space representation during the reinforcement learning process. The disturbance noise in this case is based on gSDE, and the further "interval-hold" mechanism makes the noise time-correlated. The authors conduct experiments on both classic pybullet benchmarks and musculoskeletal control environments. They show in most cases, the new exploration method can give faster reward learning compared to other baselines (SAC, gSDE-SAC, PPO, gSDE-PPO).

**Strengths:**

1. The paper is clearly written and easy to read.
2. The idea of injecting time-correlated noise is simple, novel, yet effective.
3. Extensive results comparison on a wide range of control tasks.

**Weaknesses:**

1. Lack of exploration algorithms comparison: the authors mention many exploring techniques [20-37] in the related work section, but have not compared to any of those, nor add on top of those techniques to boost the performance.
2. Lack of RL algorithms implemented: I will be eager to see how the proposed latent exploration idea can work on other RL algorithms (TRPO, TD3, etc).
3. Lack of ablation studies to provide rationale about: (1) why learn noise magnitude rather than a fixed noise magnitude, (2) the combination of the clipping parameters


**Questions:**

1. In Half cheetah experiment, the standard deviation for the reward curves is much smaller compared to the rest benchmarks. What could be the potential reason for it?
2. It seems like the proposed algorithm tends to introduce more uncertainty in the reward learning (an observation from Fig. 3-4) in most cases. Does that mean it is hard to get a very stable training from the proposed latent exploration?
3. A bit curious about the 10,000 hours of training. How many GPUs do you use? Do you think there is a way to reduce the workload to make it more friendly for normal AI labs in universities to conduct this kind of research?

**Limitations:**

The authors have well discussed the limitations in section 7. For some of the limitations (training overhead), the authors put some thought and suggest new promising work directions.

---

> ### Author Rebuttal · Authors · 2023-08-08
>
> We thank the reviewer for their comments and score.
>
> *Lack of exploration algorithms comparison: the authors mention many exploring techniques [20-37] in the related work section, but have not compared to any of those, nor add on top of those techniques to boost the performance.*
>
> We compared Lattice with two forms of action noise, standard “white” gaussian noise and state-dependent, time-correlated action noise (gSDE), because Lattice is proposed as an improvement over them. In fact, any Reinforcement Learning algorithm using white noise or gSDE can swap it with Lattice, keeping all the other implementation details unmodified. Methods such as DEP-RL [18] and curiosity-driven exploration [30], which we cite in the literature review, make use of a separate exploratory policy [18] or modifying the reward function [30, 31, 33] to promote exploration. However, both these methods still use action noise (specifically: white noise), which we argue makes them a complementary approach to Lattice, rather than a baseline. The same consideration is valid for curriculum learning [34, 35, 36, 37], where individual tasks necessitate local exploration perturbing the policy. A combination of such exploration methods with Lattice action noise is an interesting direction for future research. The paper “Pink noise is all you need” [11] (ICLR 2023), instead, proposes an alternative action noise, which is often better than Ornstein-Uhlenbeck (OU) noise. In the global answer and in the additional pdf (Figure 1), we report the results we have obtained with SAC + pink noise in the 5 PyBullet environments considered in the paper. The results corroborate our claim that correlation across action components, the distinguishing trait of Lattice, can provide a boost in performance, improving exploration in a different way from existing techniques to add correlated noise.
>
> *Lack of RL algorithms implemented: I will be eager to see how the proposed latent exploration idea can work on other RL algorithms (TRPO, TD3, etc).*
>
> We agree with the reviewer that it would be interesting to test Lattice with other RL algorithms, such as TRPO and TD3, since Lattice can be implemented with all of them. Because of the high computational load of the experiments presented in this paper, we were forced to make some decisions regarding which experiments to run.
> For the on-policy algorithm, we chose PPO because of its popularity. For off-policy algorithms, TD3 is a valid alternative to SAC. In the considered PyBullet environments, TD3 tends to perform slightly worse than SAC [1], and was found to generally perform worse than SAC with time-correlated noise [2].
>
> *Lack of ablation studies to provide rationale about: (1) why learn noise magnitude rather than a fixed noise magnitude, (2) the combination of the clipping parameters*
>
> The choice of only considering a learnt noise magnitude is related to the previously outlined experimental setup, involving only PPO and SAC. Both PPO and SAC learn a stochastic policy and, in popular implementations of these two algorithms, the noise magnitude is learnt [1]. We did not modify this characteristic of the implementation.
> The clipping from below was added to prevent the singularity of the covariance matrix (see Appendix A4). The upper limit to prevent the divergence of the covariance matrix. In the experiments we performed, the upper limit was not reached, as the variance decreases over the training. We still decided to include it in the implementation, in case future work requires limiting the magnitude of the variance.
>
> *In Half cheetah experiment, the standard deviation for the reward curves is much smaller compared to the rest benchmarks. What could be the potential reason for it?*
>
> Half-cheetah, differently from the other locomotion environments of PyBullet, is statically stable. This means that episode length is mostly 1000 (the maximum) since the beginning of the training, and episodes rarely last fewer steps. In the other environments, a large fraction of the reward variability is caused by the variable episode length in those environments where the agent needs to actively stabilize.
>
> *It seems like the proposed algorithm tends to introduce more uncertainty in the reward learning (an observation from Fig. 3-4) in most cases. Does that mean it is hard to get a very stable training from the proposed latent exploration?*
>
> We believe that the uncertainty is caused by the higher variance in behavior induced by Lattice (as showcased in the motivating example), which in some cases makes the learning curve non-monotonically increasing. We would like to remark that we have not found a significantly different variability in the final performance across random seeds.
>
> *A bit curious about the 10,000 hours of training. How many GPUs do you use? Do you think there is a way to reduce the workload to make it more friendly for normal AI labs in universities to conduct this kind of research?*
>
> The experiments were performed over several months. We dedicated 3 computers to the experiments over the development of the project. For the final experiments, we scaled the simulations to a GPU cluster. We recognize that the computation employed for this project is considerable. Most computational resources were used for the musculoskeletal simulations (Approx. 90% MyoSuite vs 10% PyBullet). One limitation of the environments used in this project is that they do not take advantage of the GPU for acceleration. Therefore, the CPU was used for the physics simulation and the GPU for the policy updates. The CPU was the bottleneck. Future RL research might greatly benefit from environments such as IsaacGym, which can exploit GPU acceleration and reduce the simulation time.
>
> [1] Raffin, A., Hill, A., Ernestus, M., Gleave, A., Kanervisto, A., & Dormann, N. (2019). Stable baselines3
>
> [2] Eberhard, Onno, et al. "Pink noise is all you need: Colored noise exploration in deep reinforcement learning." ICLR 2023

---

> > ### Comment · Reviewer_p1VW · 2023-08-15
> >
> > Thanks for your response. I would like to keep my rating unchanged.

---

### Official Review · Reviewer_mQQQ · 2023-07-28

**Soundness:** 4 excellent
**Presentation:** 4 excellent
**Contribution:** 3 good
**Rating:** 7
**Confidence:** 2

**Summary:**

The paper proposes adding time- and actuator-correlated noise instead of simple gaussian noise to improve exploration in RL.


**Strengths:**

1. The paper is well written and easy to read.
2. The approach can be easily applied to any RL algorithm. And can be thought of as a generalization of earlier work in noise for exploration (gSDE).
3. The improvements on 4 out of 8 tasks in MyoSuite is encouraging.


**Weaknesses:**

1. Is there any intuition as to which tasks are suitable for your method? It appears completely random that PPO outperforms everyone on handpose which is (I expect), an implicit subgoal in tasks like Baoding and Pen. Hence it is unclear why lattice underperforms on hand pose, elbow pose.
2. A nitpick: “ought” is an ambiguous word to use in the abstract and should be replaced.
3. Another nitpick: the arrows in the energy saving vs reward gain plot is not very clear. What do they indicate?

**Questions:**

Please see weaknesses.

**Limitations:**

Yes

---

> ### Author Rebuttal · Authors · 2023-08-08
>
> We thank the reviewer for their comments and score.
>
> *Is there any intuition as to which tasks are suitable for your method? It appears completely random that PPO outperforms everyone on handpose which is (I expect), an implicit subgoal in tasks like Baoding and Pen. Hence it is unclear why lattice underperforms on hand pose, elbow pose.*
>
> We believe there is an underlying logic, why Lattice performs strongly on the Humanoid and the object manipulation tasks. Lattice takes advantage of learnt correlations across actuators to direct the exploration towards task-relevant dimensions. Compared to Hand Reach, Baoding, Pen and Reorient, which use the same hand model (and therefore action space) as Hand Pose, we argue that Hand Pose allows fewer possibilities of exploiting “motor redundancies”. In fact, the Hand Pose task demands the agent to reach (many) target poses and thus, for each target pose many target angles with all the hand joints (therefore: 23 target values). This task does not allow learning movements involving a limited number of muscles moving always in the same way, like, e.g., baoding, where the balls always need to perform a rotation, without compromising the reward. For this reason, Lattice is not suitable for this task, as it is also not better than standard noise in Elbow Pose and Finger Pose. An empirical rule to decide whether to use Lattice vs uncorrelated noise is to compare the size of the action space with the size of the expected “action dimensionality” (Additional pdf: Figure 2).
>
> *A nitpick: “ought” is an ambiguous word to use in the abstract and should be replaced.*
>
> We have rephrased the sentence as follows:
> While this unstructured exploration has proven successful in numerous tasks, it can be suboptimal for overactuated systems.
>
> *Another nitpick: the arrows in the energy saving vs reward gain plot is not very clear. What do they indicate?*
>
> For clarity of presentation, we indicate that the 1st quadrant of the cartesian graph indicates an improvement in reward and energy saving of Lattice-PPO over PPO, while the 3rd quadrant a deterioration. We believed that it would not be clear at first glance that “higher is better” in a metric related to energy, hence we added the two arrows with the sole purpose of making the interpretation of the graph more obvious.

---

> > ### Comment · Reviewer_mQQQ · 2023-08-10
> > **Thank you for the clarifications**
> >
> > My questions have been answered. I've also increased my score to 7.

---

### Official Review · Reviewer_9Q2U · 2023-07-31

**Soundness:** 4 excellent
**Presentation:** 3 good
**Contribution:** 3 good
**Rating:** 6
**Confidence:** 3

**Summary:**

The paper presents an extension to the previously introduced RL exploration method gSDE called Lattice. The main idea is to add noise to the full covariance matrix (instead of only onto the diagonal) of the multivariate Gaussian used for exploration. In particular, the authors multiply state dependent noise with the weights of the last (linear) noise layer of the policy network (so that the resulting action probability distribution is known) and add this term to the weights and action dependent noise (gSDE). In this manner, the authors achieve correlation between action components. Both, the action and state dependent noise terms are sampled periodically, where the period determines the time-correlation in the noise terms.

**Strengths:**

- thorough experimental validation on classical RL benchmarks and high dimensional muscle RL benchmarks (myosuite)
- detailed implementation details and discussion about how Lattice works
- paper is easy to read and understand

**Weaknesses:**

- no comparison to other exploration methods such as [11], [18] or [30]
- contribution seems rather empirical (no theoretical contribution found)
- Lattice results for classical RL benchmarks (ant, hopper, walker, half cheetah, humanoid) not very convincing even in comparison to previous method (gSDE).

**Questions:**

Please address the points in the weakness section.

**Limitations:**

The authors appropriately addressed the limitiations of their work.

---

> ### Author Rebuttal · Authors · 2023-08-08
>
> We thank the reviewer for the precise summary and for praising the clarity of our exposition. We would like to address their concerns:
>
> *no comparison to other exploration methods such as [11], [18] or [30]*
>
> In our manuscript we have compared Lattice to SAC/PPO with two forms of action noise, standard “white” gaussian noise and state-dependent, time-correlated action noise (gSDE), because Lattice is proposed as an improvement over them. In fact, any Reinforcement Learning algorithm using white noise or gSDE can simply swap this action noise with Lattice, keeping all the other implementation details unmodified.
> Methods such as DEP-RL [18] and curiosity-driven exploration [30], which we cite in the related work section, make use of a separate exploratory policy [18] or curiosity as an intrinsic reward [30] to promote exploration. However, both these methods are paired with some form of action noise (specifically: white noise), which we argue makes them a complementary approach to Lattice, rather than a baseline. A combination of such exploration methods with Lattice action noise is an interesting direction for future research.
> The very recent paper “Pink noise is all you need” [11] (ICLR 2023), instead, proposes an alternative action noise. Indeed, [11] shows that pink noise is often better than Ornstein-Uhlenbeck (OU) noise (which, however, was already outperformed by gSDE [1]). Pink noise, as formulated by the authors of [11], is limited to off-policy RL. Therefore, we compared its performance with that of Lattice-SAC and not with Lattice-PPO. Here we report the results we have obtained with SAC + pink noise in the 5 PyBullet environments considered in the paper (Ant, Walker, Hopper, Half Cheetah and Humanoid, 5 seeds per environment):
>
> |                            | Ant        | Hopper     | Walker     | Half cheetah   | Humanoid   |
> |:---------------------------|:-----------|:-----------|:-----------|:---------------|:-----------|
> | SAC                        | 3381 ± 30  | 2417 ± 106 | 2741 ± 81  | 2934 ± 27      | 2122 ± 169 |
> | SAC-gSDE episode           | 3796 ± 48  | 2472 ± 64  | 2822 ± 32  | **3081 ± 93**      | 2460 ± 160 |
> | SAC-pink                   | 3648 ± 53  | 2438 ± 73  | 2580 ± 164 | 2999 ± 71      | 2116 ± 176 |
> | SAC-Lattice (ours)         | 3544 ± 212 | **2610 ± 78**  | 2718 ± 92  | 2900 ± 67      | **2742 ± 77**  |
> | SAC-Lattice episode (ours) | 3845 ± 86  | **2586 ± 23**  | 2661 ± 130 | 2948 ± 24      | **2901 ± 211** |
>
> We highlighted the results when the confidence intervals of a Lattice experiment and the non-Lattice ones are disjoint (and vice-versa). The results and the learning curves are also included in the additional 1-page pdf. Overall, we find that pink noise is a good alternative to white noise, as it produces strong results. However, as it is also a method to introduce time correlation, its performance does not improve over gSDE. We believe that these experiments corroborate our claim that correlation across action components, the distinguishing trait of Lattice, can provide a boost in performance in high-dimensional environments, improving exploration in a different way from existing techniques to add correlated noise.
>
> *contribution seems rather empirical (no theoretical contribution found)*
>
> We partially agree with the reviewer’s assessment, the results of this paper are mostly experimental. However, we believe the mathematical formulation of non-diagonal, state-dependent action noise for on-policy and off-policy RL algorithms is also of interest for the RL community, as the analytical derivation of the probability distribution (Appendix A1), of the noise rescaling factor (Appendix A2) and on the conditions to prevent the singularity of the covariance matrix (Appendix A3) were important to justify our implementation.
> We also want to point the reviewer to the analytical derivation of the effective variance for the reduced arm model, which is of course just a toy model (Appendix A1).
>
> *Lattice results for classical RL benchmarks (ant, hopper, walker, half cheetah, humanoid) not very convincing even in comparison to previous method (gSDE).*
>
> We agree with the reviewer’s assessment that the performance of gSDE is as strong as that of Lattice in most PyBullet environments, with the notable exception of the Humanoid. In the manuscript, we argue that these results should not surprise the reader, as especially in the four simpler, somewhat low-dimensional locomotion environments (Ant, Hopper, Walker and Half Cheetah) we do not see a motivation why correlated action noise should be preferable. These simpler locomotion environments do not have a large observation/action space and all the motors can contribute to the forward motion, so that the redundancy is limited. We included all environments to prove that Lattice provides strong and reliable results also in environments where action correlation does not improve performance. In the Humanoid environment, where the motor redundancy is high, Lattice substantially outperforms white noise, pink noise and gSDE. We believe that it is state of the art in SB3.
> Importantly, we demonstrate that for high-dimensional, redundant tasks Lattice achieves excellent performance (e.g. > 25 degrees of freedom). Specifically, for that we focussed on the benchmarking tasks of the MyoSuite, where Lattice reaches strong results.

---

> > ### Comment · Reviewer_9Q2U · 2023-08-17
> >
> > Thank you for the detailed and clear response. I agree with most of the points and have, hence, raised my score from 4 to 6.

---

### Author Rebuttal · Authors · 2023-08-09

This paper proposes a simple and easy-to-implement idea for exploration in reinforcement learning by exploiting the implicit knowledge in the weights of the policy network to drive exploration.  We thank all reviewers for their assessment. We think the quality of our paper improved due to your comments. We provide specific answers for each reviewer and, here, raise two topics here that were mentioned by at least two reviewers.

Additional baselines: We assessed the capability of Lattice by comparing it to a new baseline exploration noise, pink noise (just published in ICLR by Eberhard et al.), which has been shown to be effective in combination with SAC for continuous control tasks (we report the updated learning curves in Figure 1 of the additional PDF). Pink-SAC’s performance is similar to that of gSDE-SAC. However, the correlation across actuators introduced by Lattice boosts performance for the Humanoid and, to a lesser extent, for the Hopper, where it outperforms standard SAC, SAC with pink noise and SAC with gSDE. We report the table with the test performance of SAC, Pink-SAC, time-independent Lattice-SAC and episodic gSDE-SAC and Lattice-SAC:
|                            | Ant        | Hopper     | Walker     | Half cheetah   | Humanoid   |
|:---------------------------|:-----------|:-----------|:-----------|:---------------|:-----------|
| SAC                        | 3381 ± 30  | 2417 ± 106 | 2741 ± 81  | 2934 ± 27      | 2122 ± 169 |
| gSDE-SAC episode           | 3796 ± 48  | 2472 ± 64  | 2822 ± 32  | **3081 ± 93**      | 2460 ± 160 |
| Pink-SAC                  | 3648 ± 53  | 2438 ± 73  | 2580 ± 164 | 2999 ± 71      | 2116 ± 176 |
| Lattice-SAC (ours)         | 3544 ± 212 | **2610 ± 78**  | 2718 ± 92  | 2900 ± 67      | **2742 ± 77**  |
| Lattice-SAC episode (ours) | 3845 ± 86  | **2586 ± 23**  | 2661 ± 130 | 2948 ± 24      | **2901 ± 211** |

Why does Lattice not improve the results in all the 5 locomotion tasks, but just shows a clear performance increase in the Humanoid?
As suggested by the design of Lattice, we hypothesize that it can improve over baselines when high-dimensional morphologies with redundant degrees of freedom should be controlled (e.g. musculoskeletal systems, or the humanoid). We corroborate this in a novel analysis. We computed the effective (linearized) dimensionality of the learnt policies with respect to the size of the action space (additional pdf: Figure 2). We did this by evaluating the 5 policies trained in each environment for 100 test episodes. We ran the principal component analysis of the actions and computed the cumulative explained variance. We defined the effective dimensionality of a policy as the number of principal components necessary to achieve 90% explained variance.  Lattice tends to discover lower dimensional policies, especially for larger action spaces, which proves to be beneficial for the reward and the energy saving, whereas it performs as good as PPO/SAC when the dimensionality of the actions is low, or when the target space requires to control independently almost all joints. Importantly, Lattice is on par in those cases, so that it could be used as a drop-in replacement.

Overall, our work pioneers the study of exploratory noise with correlation across actuators in Reinforcement Learning. The experiments with Lattice show for the first time that biasing the action noise with the motor synergies, learnt by the policy itself, leads to specific characteristics of the control policy. The policies trained with Lattice have lower effective dimensionality, meaning that they output actions which can be embedded in a lower-dimensional subspace than the actions output by a policy trained with white or time-only correlated noise. This effect is consistent across many different environments. In those environments with task-irrelevant degrees of freedom (Humanoid) and in the presence of overactuation (musculoskeletal systems), especially in object manipulation environments, this results in higher learning efficiency, higher peak performance and lower energy cost.

---

### Comment · Area_Chair_Xm5B · 2023-08-15
**Please read and respond to authors' rebuttals**

Dear reviewers,

Thank you for your reviews. The authors have posted their rebuttal. If you have not yet done so, please read the rebuttal and the other reviews, and comment on whether the rebuttal has addressed your comments or concerns.

---

### Decision · Program_Chairs · 2023-09-21

**Decision:**

Accept (poster)

**Comment:**

This paper proposes to improve the action space exploration of RL by perturbing the network's activations, modeled as a multivariate Gaussian distribution with a full covariance matrix. It showed that this more structural exploration can lead to better learning in simulated locomotion tasks. All the reviewers agree that the paper is clearly written, and the method is simple yet effective. The main concerns raised in the reviews are lack of comparison with other exploration methods, need more RL baselines, and limited improvement in some of the evaluations. The authors' rebuttal has sufficiently addressed these concerns. All the reviewers unanimously voted for acceptance after the rebuttal and discussion. Please incorporate the improvement in the rebuttal and additional promised results into the final version of the paper.